# UNDERSTANDING THE ROLE OF POSITIONAL ENCODINGS IN SENTENCE REPRESENTATIONS

## ABSTRACT

Positional Encodings (PEs) are used to inject word-order information into transformer-based language models. While they can significantly enhance the quality of sentence representations, their specific contribution to language models are not fully understood, especially given recent findings that building natural-language understanding from language models with positional encodings are insensitive to word order. In this work, we conduct more in-depth and systematic studies of positional encodings, thus complementing existing work in four aspects: (1) We uncover the core function of PEs by identifying two common properties, Locality and Symmetry; (2) We first point out a potential weakness of current PEs by introducing two new probing tasks of word swap; (3) We first investigate the linguistic capability of PEs;(4) Based on these findings, we propose a simplified method to inject positional information into such models. Empirical studies demonstrate that this method can improve the performance of the BERT-based model on 10 downstream datasets. We hope these new probing results and findings can shed light on how to design and inject positional encodings into language models.

## 1 INTRODUCTION

Transformer-based language models with Positional Encodings (PEs) can improve performance considerably across a wide range of natural language understanding tasks. Existing work resort to either fixed (Vaswani et al., 2017; Su et al., 2021; Press et al., 2021a) or learned (Shaw et al., 2018; Devlin et al., 2019; Wang et al., 2019) PEs to infuse order information into attention-based models. To understand how PEs capture word order, prior studies apply visualized (Wang & Chen, 2020) and quantitative analyses (Wang et al., 2020) to various PEs, and their findings conclude that all encodings, both human-designed and learned, exhibit a consistent behavior: First, the position-wise weight matrices show that non-zero values gather on local adjacent positions. Second, the matrices are highly symmetrical, as shown in Figure 1. These are intriguing phenomena, with reasons not well understood.

To bridge this gap, we strive to uncover the core properties of PEs by introducing two quantitative metrics, *Locality* and *Symmetry*. Our empirical studies demonstrate that these two properties are correlated with sentence representation capability. This explains why fixed encodings are designed to satisfy them and learned encodings are favorable to be local and symmetrical. Moreover, we show that if BERT is initialized with PEs that already share good locality and symmetry, it can obtain better inductive bias and significant improvements across 10 downstream tasks.

Although PEs with locality and symmetry can achieve promising results on natural language understanding tasks (such as GLUE Wang et al. (2018)), the symmetry property itself has an obvious weakness, which is not revealed by previous work. Existing studies use shuffled text to probe the sensitivity of PEs to word orders (Yang et al., 2019a; Pham et al., 2021; Sinha et al., 2021; Gupta et al., 2021; Abdou et al., 2022), and they all assume that the meaning of sentences with random swaps remains unchanged. However, the random shuffling of words may change the semantics of the original sentence and thus cause the change of labels. For example, the sentence pair below from SNLI (Bowman et al., 2015) satisfies the entailment relation:

    a. *A man playing an electric guitar on stage*           b. *A man playing guitar on stage*

If we change the word order of the premise sentence so that it becomes "*an electric guitar playing a man on stage*", but a fine-tuned BERT still finds that the premise entails the hypothesis. Starting from this point, we design two new probing tasks of word swap: *Constituency Shuffling* and *Semantic Role Shuffling*. The former attempt to preserve the original semantics of the sentence by swapping words inside constituents (local structure) while the latter intentionally changes the semantics by swapping the semantic roles in a sentence (global structure), e.g., the agent and patient. Our probing results show that existing language models with various PEs are robust against local swaps but extremely fragile against global swaps.

Moreover, we investigate the linguistic roles of positional encodings, which have not yet been studied by prior work. Our empirical results show that there is a clear distinct role between the positional and contextual encodings in sentence comprehension: positional encodings play more of a role at the syntactic level while contextual encodings serve more at the semantic level (if the semantic task does not require word order information), and the combination of the two can consistently yield better performances on these probing tasks. As for the dependency relations, positional weights capture more short-distance dependencies while contextual weights capture more long-distance ones. Finally, based on our new findings, we propose a new method to combine positional and contextual features, which is a simple yet effective way to inject positional encodings into language models. Experimental results show that our proposed method can bring improvements across 10 sentence-level downstream tasks.

The key contributions of our work are:

- We introduce two quantitative metrics, locality and symmetry, to systematically uncover the main functions of positional encodings.

- We design two new probing tasks of word swaps, which show a weakness of existing positional encodings, namely the insensitivity against the swap of semantic roles.

- We first probe the linguistic roles of positional encodings, which reveals contextual and positional encodings play distinct roles at the syntactic level.

- Based on our findings, we propose a novel way to combine positional and contextual encodings, which can bring performance improvement without introducing complexity.

## 2 PRELIMINARIES

The central building block of transformer architectures is the self-attention mechanism (Vaswani et al., 2017). Given an input sentence: $\mathbf{X} = \{\mathbf{x}_1, \mathbf{x}_2, ..., \mathbf{x}_n\} \in \mathbb{R}^{n \times d}$, where $n$ is the number of words and $d$ is the dimension of word embeddings, then the attention computes the output of the $i$-th token in this way:

$$\bar{\mathbf{x}}_i = \sum_{j=1}^{n} \frac{\exp(\alpha_{i,j})}{Z} \mathbf{x}_j \mathbf{W}^V \quad \text{where } \alpha_{i,j} = \frac{(\mathbf{x}_i \mathbf{W}^Q)(\mathbf{x}_j \mathbf{W}^K)^\mathsf{T}}{\sqrt{d}}, Z = \sum_{j=1}^{n} \exp(\alpha_{i,j}) \quad (1)$$

Self-attention heads do not intrinsically capture the word orders in a sequence. Therefore, specific methods are used to infuse positional information into self-attention Dufter et al. (2022).

**Absolute Positional Encoding (APE)** computes a positional encoding for each token and add it to the input content embedding to inject position information in the original sequence. The $\alpha_{i,j}$ in Equation 1 are then written:

$$\alpha_{i,j} = \frac{(\mathbf{x}_i + \mathbf{p}_i)\mathbf{W}^Q ((\mathbf{x}_j + \mathbf{p}_j)\mathbf{W}^K)^\mathsf{T}}{\sqrt{d}} \quad (2)$$

where $\mathbf{p}_i \in \mathbb{R}^d$ is a position embedding of the $i$th token, obtained by **fixed** (Vaswani et al., 2017; Dehghani et al., 2018; Takase & Okazaki, 2019; Shiv & Quirk, 2019; Su et al., 2021) or **learned** encodings (Gehring et al., 2017; Devlin et al., 2019; Wang et al., 2019; Press et al., 2021b). Further, TUPE model simplifies Equation 2 by removing two redundant items (see details in Section A of the appendix):

$$\alpha_{i,j} = \frac{(\mathbf{x}_i \mathbf{W}^Q)(\mathbf{x}_j \mathbf{W}^K)^\mathsf{T} + (\mathbf{p}_i \mathbf{U}^Q)(\mathbf{p}_j \mathbf{U}^K)^\mathsf{T}}{\sqrt{d}} \quad (3)$$

**Relative Positional Encoding (RPE)** produces a vector $\mathbf{r}_{i,j}$ or a scalar value $\beta_{i,j}$ that depends on the relative distance of tokens. Specifically, these methods apply such vector or bias to the attention head so that the corresponding attentional weight can be updated based on the relative distance of two tokens (Shaw et al., 2018; Raffel et al., 2019):

$$\alpha_{i,j} = \frac{\mathbf{x}_i\mathbf{W}^Q(\mathbf{x}_j\mathbf{W}^K + \mathbf{r}_{i,j}^K)^\mathsf{T}}{\sqrt{d}} \quad ; \quad \alpha_{i,j} = \frac{(\mathbf{x}_i\mathbf{W}^Q)(\mathbf{x}_j\mathbf{W}^K)^\mathsf{T} + \beta_{i,j}}{\sqrt{d}} \tag{4}$$

where the left mode uses a vector $\mathbf{r}_{i,j}$ while the right uses a scalar value $\beta_{i,j}$, for infusing relative distance into attentional weight.

Recent research of RPEs has been remarkably vibrant, with the emergence of diverse novel and promising variants (Dai et al., 2019; He et al., 2020; Press et al., 2021a).

**Unified Positional Encoding.** Inspired by TUPE (Ke et al., 2021), we rewrite all above absolute and relative positional encodings in a unified way:

$$\alpha_{i,j} = \frac{\overbrace{\gamma_{i,j}}^{contextual} + \overbrace{\delta_{i,j}}^{positional}}{\sqrt{d}} \tag{5}$$

where, the left half of the numerator, $\gamma_{i,j}$, captures contextual correlations (or weights), i.e., the semantic relations between token $x_i$ and $x_j$. In this case, it is $\gamma_{i,j} = (\mathbf{x}_i\mathbf{W}^Q)(\mathbf{x}_j\mathbf{W}^K)^\mathsf{T}$. $\delta$, the right half, captures positional correlations, i.e., the positional relations between tokens $x_i$ and $x_j$. For example, TUPE's positional correlation can be represented as $\delta_{i,j} = (\mathbf{p}_i\mathbf{W}^Q)(\mathbf{p}_j\mathbf{W}^K)^\mathsf{T}$ while the relative encoding in Shaw et al. (2018) can be represented as $\delta_{i,j} = \mathbf{x}_i\mathbf{W}^Q(\mathbf{r}_{i,j}^K)^\mathsf{T}$. Thus, existing positional encodings all add contextual and positional correlations together in every attention head.

## 3 POSITIONAL ENCODINGS ENFORCE LOCALITY AND SYMMETRY

### 3.1 THE PROPERTIES OF LOCALITY AND SYMMETRY

Existing work analyze positional encodings with the help of visualizations Wang & Chen (2020); Wang et al. (2020); Abdou et al. (2022), and their analyses of either fixed or learned encodings led to similar visualized results, as shown in Figure 1. These position-wise weight matrices are computed by using the *Identical Word Probing* proposed by Wang et al. (2020): many repeated identical words are fed to the pre-trained language model, so that the attention values ($\alpha_{i,j}$) in Equation 5 are unaffected by contextual weights (See details of visualizations in Section B.1). Each matrix in this figure is a positional weight map, where each row is a

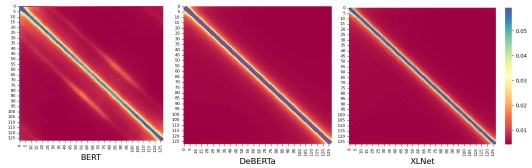

Figure 1: Visualizations of different pre-trained language models by using Identical Word Probing (Wang et al., 2020). The attention weights are averaged across different layers.

vector for the $i$-th position and the element at $(i, j)$ indicates the attention weight between the $i$-th position and the $j$-th position. We can first observe these attention matrices are all diagonal heavy, which means various positional encodings highly attend to local positions. Second, all matrices are nearly symmetrical. We call these two phenomena the *Locality* and *Symmetry* of positional encodings, and we provide a linguistic explanation for the two properties in Appendix B.3. The symmetry property has been discovered and quantified already by Wang et al. (2020). Here, we provide a more in-depth analysis of symmetry. We will also point out the potential flaw of symmetry itself, which is not considered by prior work. To better understand how encodings capture word order, we introduce two quantitative metrics to depict the Locality and Symmetry for an attentional weight vector $\epsilon_i$, where the element $\epsilon_{i,j}$ can be denoted as:

$$\epsilon_{i,j} = \frac{\exp(\alpha_{i,j})}{\sum_{j=1}^n \exp(\alpha_{i,j})} \quad \text{where } \epsilon_{i,j} \geq 0 \text{ and } \sum_{j=1}^n \epsilon_{i,j} = 1 \tag{6}$$

*Locality* is a metric that depicts the degree of the gathering of weights in local positions for an attentional weight vector. Given a weight vector for the $i$-th position $\epsilon_i = \{\epsilon_{i,1}, \epsilon_{i,2}, ..., \epsilon_{i,n}\}$, we define locality as:

$$\text{Locality}(\epsilon_i) \in [0, 1] = \sum_{j=1}^n \frac{\epsilon_{i,j}}{2^{|i-j|}} \tag{7}$$

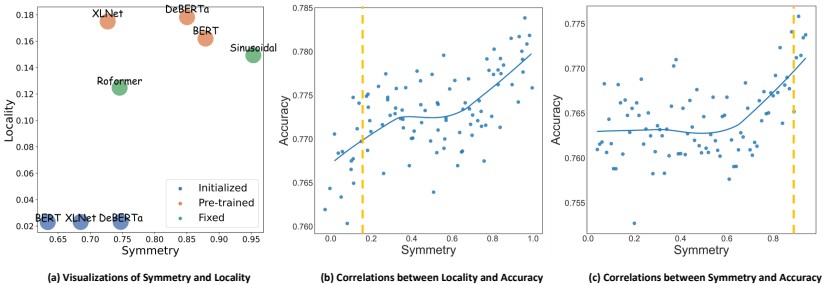

Figure 2: Empirical studies of the properties of locality and symmetry. The accuracy is tested on the MR dataset (Pang & Lee, 2005) The yellow line shows the locality or symmetry for the pre-trained BERT. Results on SUBJ dataset (Pang & Lee, 2004) are shown in Figure 7.

Here, a value of 1 means the vector perfectly satisfies the locality property. For example, given a sequence whose length is 5 and a weight vector for the first position $[1, 0, 0, 0, 0]$, the locality is 1, which means it perfectly matches the locality. In contrast, the locality is $1/16$ if the weight only attends the last position $[0, 0, 0, 0, 1]$. For measuring the locality of a matrix, we average the locality values of all vectors in the matrix.

***Symmetry*** is a metric that describes how symmetrical the weights scatter around the current position for an attentional weight vector. We adapt the *Symmetrical Discrepancy* from Wang et al. (2020) for this goal:

$$\text{Symmetry}(\epsilon_i) \in [0, 1] = 1 - \sum_{j=1}^{\lfloor n/2 \rfloor} \text{Norm}\left(\frac{|\epsilon_{i,j} - \epsilon_{i,n-j+1}|}{\lfloor n/2 \rfloor}\right) \tag{8}$$

Here, a value of 1 means that the vector is completely symmetrical. We modify the original formula in two points: First, we apply a min-max normalization to each position to obtain more uniform distributions, because the values of the original one extremely cluster around $0$. Second, we reverse the value so that 1 means a perfect symmetry instead of $0$. Likewise, the average value of all vectors in a matrix is used as the matrix-level symmetry.

## 3.2 ARE LOCALITY AND SYMMETRY LEARNED?

The manually designed encodings Sinusoidal (Vaswani et al., 2017) and Roformer (Su et al., 2021) both satisfy the symmetry and locality properties. However, it is not clear why they were designed this way. More surprisingly, learned encodings all display locality and symmetry. Therefore, one may ask whether the two properties are learned after pre-training, and what effect they have.

To answer this question, we use our two proposed metrics to quantify the positional weight matrix (the averaged weight across layers) before and after pre-training. Specifically, three language models, BERT (Devlin et al., 2019), XLNet (Yang et al., 2019b) and DeBERTa (He et al., 2020) are tested in this experiment. As shown in the left in Figure 2, the three language models all become much more local and symmetrical after pre-training, which proves that the two properties are indeed learned.

To further explain why positional encodings have a preference for learning these two properties, we probe the correlations between the two properties and the representation ability in downstream tasks. To avoid pre-training all language models from scratch, we use static word embeddings from GloVe (Pennington et al., 2014) and an encoder that is fully based on our handcrafted positional encodings for a sentence classification task. The benefit is that we can adjust the hyper-parameter in the handcrafted encodings to obtain encodings with different degrees of locality and symmetry, so that we can evaluate the correlations precisely. Specifically, we obtain around 100 encoders whose locality (or symmetry) varies from 0.01 to 1.0 and test their accuracy on the MR sentiment analysis task. We will describe our handcrafted encodings in Section 3.3. The details of the encoder used in this experiment is described in Appendix B.4.

The middle figure in Figure 2 shows the results for different locality values. In this experiment, the symmetry value is 1.0 for all encoders. We observe that the accuracy constantly increases as the locality of encodings strengthens, which means a higher locality induces better sentence representation. The yellow line is the locality value for BERT (around 0.2), and BERT actually does not

| Model | Size | Sentiment Analysis | | | Textual Entailment | | | Paraphrase Identification | | Textual Similarity | | Avg |
|---|---|---|---|---|---|---|---|---|---|---|---|---|
| | | MR (22K) | SUBJ (20K) | SST-2 (68.8K) | QNLI (110K) | RTE (5.5K) | MNLI (413K) | MRPC (5.4K) | QQP (755k) | STS-B (8.4K) | SICK-R (9.4K) | |
| BERT | 110M | $72.5_{\pm5.3}$ | $91.0_{\pm2.7}$ | $86.4_{\pm2.7}$ | $85.8_{\pm1.0}$ | $59.2_{\pm1.2}$ | $78.2_{\pm0.8}$ | $73.5_{\pm1.8}$ | $88.7_{\pm0.6}$ | $77.8_{\pm4.1}$ | $64.9_{\pm6.0}$ | 77.8 |
| BERT-$A^*$-$s$ | 113M | $79.4_{\pm2.9}$ | $93.7_{\pm0.6}$ | $88.0_{\pm0.7}$ | $86.3_{\pm1.1}$ | $59.4_{\pm2.7}$ | $78.8_{\pm0.4}$ | $81.5_{\pm2.2}$ | $88.7_{\pm0.4}$ | $83.6_{\pm2.0}$ | $76.3_{\pm1.1}$ | 81.6 |
| BERT-$A^*$ | 138M | $78.2_{\pm3.5}$ | $93.0_{\pm0.8}$ | $88.1_{\pm1.0}$ | $87.0_{\pm0.5}$ | $61.0_{\pm1.4}$ | $78.9_{\pm0.9}$ | $80.9_{\pm3.9}$ | $89.2_{\pm0.3}$ | $84.3_{\pm2.5}$ | $76.0_{\pm4.7}$ | 81.7 |

Table 1: Evaluations of handcrafted encodings across 10 downstream tasks. We report the average score (Spearman correlation for textual similarity and accuracy for others) of five runs using different learning rates. $^*$ means the encodings are learnable and $s$ means that positional encodings are shared within the attention headers of layers.

have an extreme locality. Experimental results on another dataset (Figure 7) show that the accuracy growth slows down at a particular locality value (0.3), which means that a perfect locality is unnecessary. The right figure in Figure 2 shows the results for different symmetry values. In this experiment, we vary the symmetry while keeping the locality in the interval $[0.15, 0.3]$, which is close to the value of BERT. Because the change of symmetry will impact the value of locality, we can only observe this type of partial correlation. We find that symmetry affects performance only after a certain value (0.65), and a better symmetry leads to better accuracy. Also, the encodings of the pre-trained BERT are highly symmetrical. Besides, experiments on SUBJ dataset (Pang & Lee, 2004) obtain similar conclusions, as shown in Figure 7.

We conclude that positional encodings with more suitable locality and symmetry can yield better performance on downstream tasks, which may explain why fixed encodings are designed to meet the two properties and why learned encodings all exhibit this behavior. However, encodings are not perfectly local, which might be due to the network architectures and the specific target tasks.

## 3.3 CAN LOCALITY AND SYMMETRY YIELD BETTER INDUCTIVE BIAS?

Given that locality and symmetry stand out as important learned features of existing positional encodings, it begs the question that what happens if a language model is initialized with positional encodings with good locality and symmetry.

For this purpose, we replace the positional correlations $\delta_{i,j}$ in Equation 5 with handcrafted Positional Encodings to probe. There are various human-designed positional encodings, e.g., sinusoidal encodings (Vaswani et al., 2017), rotary encodings (Su et al., 2021) and ALiBi (Press et al., 2021a), but the locality and symmetry cannot be modified easily for these encodings. To address this issue, we propose the *Attenuated Encoding*, which use a Gaussian kernel (Guo et al., 2019):

$$\delta_{i,j} = \Phi(l_{i,j}) = \frac{\exp(\alpha_{i,j})}{\sum_{j=1}^{n} \exp(\alpha_{i,j})} \qquad \text{where } \alpha_{i,j} = \begin{cases} -s\,w\,l_{i,j}^2 & i \le j \\ -w\,l_{i,j}^2 & i > j \end{cases} \qquad (9)$$

where $l_{i,j}$ is the relative distance, $w > 0$ is a scalar parameter that controls the locality value, and $s$ is a scalar parameter that controls the symmetry value. Note that there are two key differences between our encodings and other manually designed ones such as the T5 bias (Raffel et al., 2019) and ALiBi (Press et al., 2021a). First, the output generated by our method is an attentional vector (or a discrete probability distribution) that can be regarded as a type of attention mechanism. Thus, we can estimate the locality and symmetry individually. ALiBi biases, in contrast, cannot be measured by our proposed metrics directly. Second, we can adjust the hyper-parameters in our method for obtaining encodings with different localities and symmetry, which ALiBi does not allow.

In this experiment, we adjust the parameter $w$ and $s$ for obtaining weight vector $\delta$ that share similar locality and symmetry with pre-trained BERT (Locality=0.17 and Symmetry=1.0). After, we pre-train BERT$_{base}$ initialized with $\delta$ and compare them to learned encodings on downstream natural language understanding tasks. Three variants are compared with the original BERT: 1) BERT-$A^*$-$s$ uses learnable and shared $\delta$, but the weights are shared inside a particular layer; 2) BERT-$A^*$ uses learnable but not shared $\delta$, which means $\delta$ is different in each attentional head. More details of the datasets and pre-training are shown in the Appendix Section B.6 and Section B.5, respectively. The empirical results are shown in Table 1. We observe that both BERT-$A^*$-$s$ and BERT-$A^*$ can significantly outperform the original BERT, which demonstrates positional encodings with initialization of suitable locality and symmetry can have better inductive bias in sentence representation. Besides,

| Model | Symmetry | Locality | Original | Shuffle-3 ($\Delta$) | Shuffle-4 ($\Delta$) | Shuffle-5 ($\Delta$) | Shuffle-6 ($\Delta$) | Random ($\Delta$) | Original | Shuffle-SR ($\Delta$) |
|---|---|---|---|---|---|---|---|---|---|---|
| BERT | 87.9 | 16.2 | 89.8 | -0.4 | -0.6 | -0.3 | -0.5 | -2.7 | 89.8 | -63.9 |
| ALBERT | 82.0 | 20.3 | 91.8 | -0.5 | -1.1 | -1.3 | -1.7 | -6.0 | 92.0 | -66.8 |
| DeBERTa | 85.0 | 17.8 | 91.6 | -0.5 | -0.7 | -1.3 | -1.1 | -5.1 | 91.6 | -58.9 |
| XLNet | 72.7 | 17.5 | 91.5 | -0.2 | -0.3 | -0.7 | -1.2 | -5.4 | 91.3 | -57.8 |
| StrucBERT | 96.3 | 7.5 | 90.9 | -0.5 | -0.9 | -1.3 | -1.1 | -4.4 | 90.8 | -44.6 |

Table 2: Results of Constituency Shuffling and Semantic Role Shuffling, measured by accuracy. Shuffle-$x$ means phrases with length $x$ are shuffled. Shuffle-SR means the semantic roles of agent and patient are swapped.

the visualizations of attentional heads in BERT are shown in Figure 8 and the visualizations of $\delta$ are shown in Figure 9. In fact, there is a great diversity of behaviors within different attentional heads.

### 3.4    WHAT IS THE DRAWBACK OF SYMMETRY?

Although positional encodings with good symmetry perform well on a series of downstream tasks, the symmetry property has an obvious flaw in sentence representations, which is ignored by prior studies. Existing probes study the sensitivity of language models to word order by shuffling the words in a sentence, and they can be roughly divided into three categories: random swap (Pham et al., 2021; Gupta et al., 2021; Abdou et al., 2022), n-gram swap (Sinha et al., 2021), and subword-level phrase swap (Clouatre et al., 2022). All these works assume that the labels of the randomly shuffled sentences are unchanged. However, this is obviously not the case. In particular, the shuffled sentence may have another label (think of the textual entailment example from the introduction).

To address the issue, we propose two new probing tasks of word swaps: *Constituency Shuffling* and *Semantic Role Shuffling*. *Constituency Shuffling* aims to disrupt the inside order of constituents, which is able to change the word order while preserving the maximum degree of original semantics.

A constituent parsing case is shown in Figure 3, and we can shuffle the word order inside some phrases, e.g., the noun phrase "*an electric guitar*" while the semantic will not be changed (the grammar structure may be destroyed). We construct different shuffled datasets by phrase length, e.g., "*an electric guitar*" is a phrase of length 3 and we can obtain tri-gram shuffled sets. Datasets constructed by constituency shuffling are referred to as Shuffle-$x$ and $x$ means the length of phrase. On the other hand, *Semantic Role Shuffling* intentionally changes the semantics by swapping the order of the agent and patient of sentences and thus results in a new sentence with different meanings. In Figure 3, "*a man*" as the entity that performs the action, technically known as the agent, and "*an electric guitar*" as the entity that is involved in or affected by the action, which is called the patient. We refer to this dataset as Shuffle-SR because it swap the semantic roles in a sentence. Some shuffled examples are shown in Table 5.

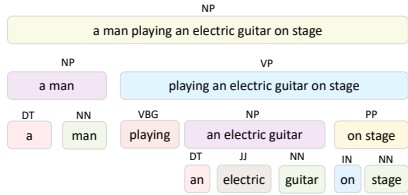

Figure 3: Illustration of constituent parsing for one sentence in SNLI "*a man playing an electric guitar on stage*". The result is generated by Berkeley Neural Parser.

The distinction of our proposed two probing tasks is that one preserves the semantics while another changes the semantics. Then, we can probe the capability of language models to correctly recognize the new sentence's meaning. Specifically, the Stanford Natural Language Inference (SNLI) (Bowman et al., 2015) dataset is used in this experiment and it provided constituent structure for each sentence. To probe the sensitivity of language models to the two types of shuffling, we fine-tune 5 pre-trained language models with good symmetry on SNLI training set and evaluate them on the newly constructed Shuffle-$x$ and Shuffle-SR datasets (see details in section B.2). The overall results of word swap probing are shown in Table 2 We first observe performances of all language models across Shuffle-$x$ sets basically do not degenerate, which confirms the benefits of the locality and symmetry properties. Second, most models fail on the Shuffle-SR dataset, which demonstrates local symmetry does not capture global position changes well, which explain the reason that BERT fails on the example: "*an electric guitar playing a man on stage*". Although the local symmetry learned by positional encodings can performs well on a series of language understanding tasks, the symmetry itself has obvious flaws. The better performance of StrucBERT on the Shuffle-SR suggests that

introducing additional order-sensitive training tasks may improve this problem. More details of the probing tasks are described in Appendix B.2

## 4 LINGUISTIC ROLES OF POSITIONAL ENCODINGS

The syntactic structure is crucial to our understanding of sentences. Many studies have shown that a substantial amount of linguistic knowledge can be found in contextual word representations, e.g., subject-verb agreement (Goldberg, 2019) and dependency tree (Hewitt & Manning, 2019; Wu et al., 2020). Follow-up work further systematically analyzed the syntactic and semantic capabilities of the BERT model (Clark et al., 2019; Jawahar et al., 2019; Lin et al., 2019; Rogers et al., 2020). Nonetheless, the linguistic roles of Positional Encodings are still under-explored. In this section, we discuss what linguistic knowledge PEs have learned.

### 4.1 LINGUISTIC PROBING TASKS

We first conduct an ablation study to check the importance of positional and contextual encodings. The experimental details are shown in Appendix (Section C.2), and we find that the removal of any encodings degrade the model performances (Table 7). We therefore hypothesize that both encodings play a role in sentence comprehension and have different responsibilities.

Positional and contextual weights are usually entangled in every attentional head, and therefore the behavior of positional encodings cannot be observed independently (as shown in Equation 5). To address this, we hide the contextual correlation $\gamma_{i,j}$ in Equation 5 of a pre-trained BERT (instead of removing the contextual encodings completely) and thus the attentional weight $\alpha_{i,j}$ only depends on positional correlation. Note that this operation does not alter the structure of the original network because a softmax layer is applied to the vector $\alpha_\mathbf{i}$, and the output is still an attentional weight vector that can be regarded as a kind of discrete probability distribution. Therefore, the output sentence representation is somewhat decoupled from contextual encoding. We refer to this adapted model as BERT-$p$. For comparison, we remove the positional correlations $\theta_{i,j}$ to obtain BERT-$c$. We do the same for a pre-trained TUPE model, to obtain TUPE-$p$ and TUPE-$c$.

In this linguistic probing, we adopt widely used 10 probing tasks (Conneau et al., 2018) with a standard evaluation toolkit (Conneau & Kiela, 2018). These tasks are a series of classification tasks that covers three categories: SentLen (Surface), WC (Surface), BShift (Syntactic), TreeDepth (Syntactic), TopConst (Syntactic), Tense (Semantic), SubjNum (Semantic), ObjNum (Semantic), SOMO (Semantic), CoordInv (Semantic). The details of this experiment are described in Section C.3.

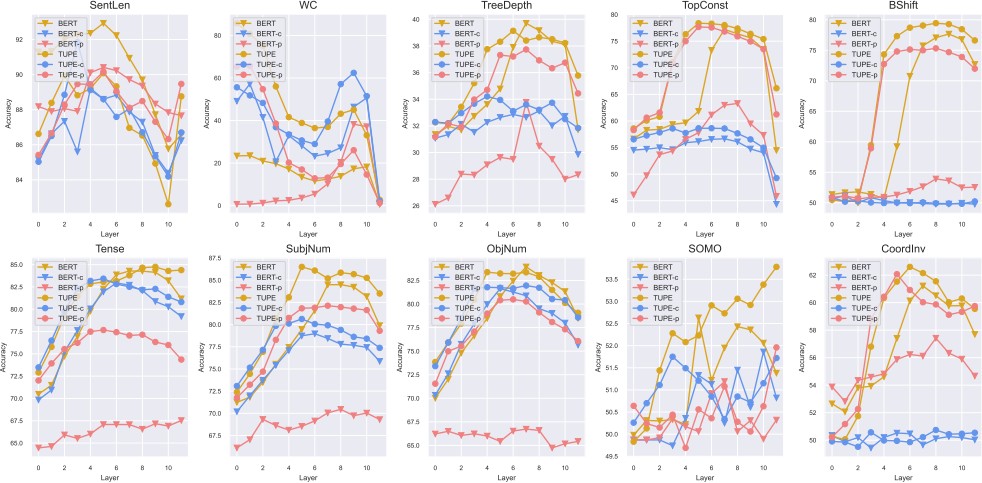

Figure 4: Results of linguistic probing tasks across different layers. BERT-based models are shown in triangle while TUPE-based models are shown in circle. The red, blue and yellow lines represent the use of positional weights, contextual weights and a combination of both, respectively.

Figure 4 gives the results. We first observe that the combination of contextual and positional encodings can have better performances across all probing tasks (yellow lines). Secondly, compared to contextual encodings, positional encodings perform better on syntactic tasks (TreeDepth, TopConst, BShift), which require more information of word orders. On semantic tasks, contextual encodings outperform contextual encodings on Tense and ObjNum while performs poorly when the semantic probing tasks require order information (CoordInv). Thirdly, a hierarchical structure exists here when we check the peak of probing tasks for each model, as observed by Jawahar et al. (2019). For surface tasks, the surface knowledge is stored more in bottom layer, syntactic knowledge is in middle layer and semantic knowledge is in middle and top layer. Therefore, we conclude that positional encodings play more of a role at the syntactic level tasks. On semantic tasks, especially position-independent ones, contextual encodings are more important.

## 4.2 DEPENDENCY SYNTAX OF POSITIONAL ENCODINGS

Clark et al. (2019) showed that particular heads of BERT indeed correspond remarkably well to particular long-distance dependencies, like ccomp and mark. To test the dependency knowledge stored in positional weights, we follow the syntactic probing test by Clark et al. (2019). Specifically, each head in PLMs is regarded as a simple predictor of dependency relations. Given the attention weight vector of an input word, we output the word with the highest values and think the pair of words hold some type of dependency relation. While no single head can perform well on all relations, the best-performed head is selected as the final ability of a model for each particular relation. In this experiment, we adopt the original TUPE that uses absolute positional encodings as our base model. The ability of contextual and positional weights is evaluated by removing the unrelated encodings in Eq 5, e.g., the first term is removed when checking the syntactic ability of PEs. The two variants are referred as to Contextual Attention (CA) and Positional Attention (PA).

We extract attention maps from BERT on the MRPC (Dolan & Brockett, 2005) annotated by dependency parser of spaCy [1]. We report the results on top-20 dependency relations. Figure 5 shows performance on different relations function of word distance in these relations (Table 8 gives relation-specific results). First, we observe that positional attention significantly more important than contextual attention in short-distance dependency relations (distance from 1 to 4). Second, contextual attention performs takes the lead on long-distance relations (after 6). Again, the combination of the two features can yield the best performance. The

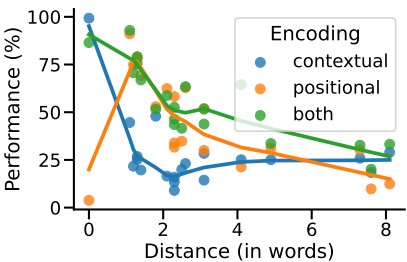

Figure 5: Accuracy of top-20 dependency relations. Detailed results in Table 8, Appendix.

"outlier" in the lower left corner is the Root dependency. Because this relation is a self-reflexive edge, contextual (or self) attentions can performs well on it while learned PEs do not attend to the current word itself, e.g., visualizations of BERT and DeBERTa in Figure 1.

## 5 HOW TO COMBINE CONTEXTUAL AND POSITIONAL FEATURES

As discussed before, the core function of positional encodings is to symmetrically combine local units. As for the linguistic role, the contextual weights are good at long-distance syntactic relations while positional weights are suitable for short dependencies. The combination of the two weights can lead to better sentence representations. The add operation of contextual and positional correlation ($\gamma_{i,j}$ and $\delta_{i,j}$ in Equation 5) is widely used in many PLMs (Devlin et al., 2019; Liu et al., 2019; Yang et al., 2019b; Ke et al., 2021), as shown in Eq 5. However, to comprehend a sentence, the semantic meaning is composed bottom-up: morpheme → word → phrase → clause → sentence. For example, the two sub-words are composed to a particular meaning {*"context"*,*"##ual"*} → *"contextual"* and the two words are composed to a phrase {*"take"*,*"off"*} → *"take off"*. When we reach word *take* and *off*, the two words should be composed of a phrase first and then its meaning is understood by using the long-distance context. At least, it is not necessary to consider contextual in-

---

[1] https://spacy.io/api/dependencyparser

| Model | Size | Sentiment Analysis | | | Textual Entailment | | | Paraphrase Identification | | Textual Similarity | | Avg |
|---|---|---|---|---|---|---|---|---|---|---|---|---|
| | | MR (22K) | SUBJ (20K) | SST-2 (68.8K) | QNLI (110K) | RTE (5.5K) | MNLI (413K) | MRPC (5.4K) | QQP (755k) | STS-B (8.4K) | SICK-R (9.4K) | |
| $A^*$ | 138M | $78.2_{\pm3.5}$ | $93.0_{\pm0.8}$ | $88.1_{\pm1.0}$ | $\mathbf{87.0}_{\pm0.5}$ | $61.0_{\pm1.4}$ | $\mathbf{78.9}_{\pm0.9}$ | $80.9_{\pm3.9}$ | $89.2_{\pm0.3}$ | $84.3_{\pm2.5}$ | $77.0_{\pm4.7}$ | 81.8 |
| $A^*$-Seq | 138M | $78.7_{\pm2.4}$ | $93.3_{\pm0.7}$ | $\mathbf{88.7}_{\pm0.3}$ | $86.5_{\pm0.4}$ | $\mathbf{64.1}_{\pm2.6}$ | $78.1_{\pm0.4}$ | $\mathbf{81.1}_{\pm2.1}$ | $89.5_{\pm0.3}$ | $\mathbf{85.1}_{\pm1.9}$ | $\mathbf{79.3}_{\pm1.1}$ | **82.5** |
| ALiBi | 110M | $78.0_{\pm4.1}$ | $90.1_{\pm2.2}$ | $84.6_{\pm1.3}$ | $84.4_{\pm0.3}$ | $63.6_{\pm3.9}$ | $74.3_{\pm0.3}$ | $76.1_{\pm2.0}$ | $87.2_{\pm0.5}$ | $82.2_{\pm4.6}$ | $76.3_{\pm4.3}$ | 79.7 |
| ALiBi-Seq | 110M | $\mathbf{78.2}_{\pm2.3}$ | $\mathbf{93.2}_{\pm1.3}$ | $\mathbf{86.1}_{\pm0.5}$ | $84.4_{\pm0.4}$ | $62.0_{\pm2.0}$ | $\mathbf{76.7}_{\pm1.3}$ | $\mathbf{76.4}_{\pm1.0}$ | $\mathbf{88.6}_{\pm0.5}$ | $82.1_{\pm1.8}$ | $75.9_{\pm3.7}$ | **80.4** |

Table 3: Evaluations of our sequence combinations of positional and contextual encodings across 10 downstream tasks. $^*$ means the encodings are learnable. We report the average score (Spearman correlation for textual similarity and accuracy for others) of five runs using different learning rates.

formation until we compose basic semantic units like words (*"contextual"*) and short phrase (*"take off"*). Therefore, a key question arises: how to combine the contextual and positional weights?

To combine the contextual and positional correlations better and according to our analysis, we propose a new strategy: **Sequence Combination**. This method uses two correlations in a particular order. First, the positional correlation $\delta_{i,j}$ in Equation 5 is used to combine nearby tokens for composing basic semantic units, e.g., $\{take,\ off\} \rightarrow take\ off$. Afterward, contextual correlation $\gamma_{i,j}$ is applied to these composing units and thus they can be better understood under the contexts. We refer to the separate usage of the two correlations as Contextual Attention and Positional Attention, respectively, and this process can be written as:

$$\bar{\mathbf{X}} = \text{CA}(\text{PA}(\mathbf{X})) \tag{10}$$

The computation of CA is obtained by removing the positional item $\delta_{i,j}$ from Equation 5, and PA can be computed through the weighted sum of the input sequence by using $\delta_{i,j}$. A code example of the implementation is shown in Listing 3.

In this experiment, we use two positional encodings as baselines: *Learned Attenuated Encodings* and *Fixed ALiBi* (Press et al., 2021a). After, the sequence combination is applied to the two baselines for comparison. Note that to make ALiBi suit for our experiment, we apply a softmax layer to the linear biases to obtain an attentional vector, and thus it can be used individually for positional attention module. The experimental results are shown in Table 3. We observe our proposed sequence combination outperforms the original add operation without introducing additional parameters, which shows that prioritizing local semantic unit composition is beneficial for sentence representation. Besides, the sequence combination can achieve lower training and validation loss than the original BERT using the same steps (Figure 6). A conclusion is that we can subtly adjust the sequence of using positional encoding without introducing additional parameters to obtain a certain degree of improvement in the language understanding tasks.

# 6 CONCLUSION

We have proposed a series of probing analyses for understanding the role of positional encodings in sentence representations. We find two main properties of existing encodings, Locality and Symmetry, which is correlated with the performance of downstream tasks. Meanwhile, we point out an obvious flaw of the symmetry property. We further complement existing research by distinguishing the linguistic roles of positional and contextual encodings, and by proposing to combine them sequentially rather than additively.

The limitations of this work are two-fold. First, our analysis is limited to the natural language understanding of the English language. Different languages display different word order properties. For instance, English is subject-verb-object order (SVO) while Japanese is subject-object-verb order (SOV), and natural language generation models are not included in this work. Besides, a recent work finds that the autoregressive models without any explicit positional encoding are still competitive with standard models (Haviv et al., 2022), which shows the generative model might not be a perfect target for researching order information. Second, although our handcrafted positional encodings satisfy the Symmetry property, it merely replicates the limitations of current positional encoding, albeit in a simplified form. Further architecture development should address the problem of the "*an electric guitar playing a man on stage.*" mentioned in the introduction.

## 7 REPRODUCIBILITY STATEMENT

The complete experimental settings, as well as the implementation details, are given in Section B. Besides, we have submitted our source code and will make it publicly available.

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

## A  DETAILS OF TUPE MODEL

In absolute positional encoding, the positional encoding is added together with the contextual encoding:

$$\alpha_{ij} = \frac{(\mathbf{x}_i + \mathbf{p}_i)\mathbf{W}^Q\big((\mathbf{x}_j + \mathbf{p}_j)\mathbf{W}^K\big)^\mathsf{T}}{\sqrt{d}} \tag{11}$$

wehre $\mathbf{p}_i \in \mathbb{R}^d$ is a position embedding of the $i$-th token. Further, the above equation can be expanded as:

$$\alpha_{ij} = \frac{(\mathbf{x}_i\mathbf{W}^Q)(\mathbf{x}_j\mathbf{W}^K)^\mathsf{T}}{\sqrt{d}} + \frac{(\mathbf{x}_i\mathbf{W}^Q)(\mathbf{p}_j\mathbf{W}^K)^\mathsf{T}}{\sqrt{d}} + \frac{(\mathbf{p}_i\mathbf{W}^Q)(\mathbf{x}_j\mathbf{W}^K)^\mathsf{T}}{\sqrt{d}} + \frac{(\mathbf{p}_i\mathbf{W}^Q)(\mathbf{p}_j\mathbf{W}^K)^\mathsf{T}}{\sqrt{d}} \tag{12}$$

There are four terms in this expression: context-to-context, context-to-position, position-to-context, and position-to-position. While the first and the fourth term are intuitive, the token encodings and positional encodings do not have strong correlations with each other, and the context-position correlations may even induce unnecessary noise. Based on this analysis, Ke et al. (2021) propose TUPE (Transformer with Untied Positional Encoding) that removes the second and third redundant terms and introduces different parameters for the position encoding:

$$\alpha_{ij} = \frac{(\mathbf{x}_i\mathbf{W}^Q)(\mathbf{x}_j\mathbf{W}^K)^\mathsf{T} + (\mathbf{p}_i\mathbf{U}^Q)(\mathbf{p}_j\mathbf{U}^K)^\mathsf{T}}{\sqrt{d}}, \tag{13}$$

Here, $\mathbf{U}^Q$ and $\mathbf{U}^K$ are weights that need to be learned, capturing positional queries and keys, respectively. Their empirical results confirm that the removal of the two context-to-position terms consistently improves the model performance on a series of tasks.

## B  DETAILS OF EXPERIMENTS

### B.1  VISUALIZATIONS OF POSITIONAL ENCODINGS

To understand what positional encodings learn after pre-training, we visualize the positional weights in attentional heads. The Identical Word Probing is adopted in this experiment (Wang et al., 2020). The used pre-trained language models are shown in Table 4, and the repeated words are randomly selected from the corresponding vocabulary. Note that sub-tokens like single characters and non-physical words are removed. For visualization, we adopt the Identical Word Probing proposed

| Model | Size | Version | Language |
|-------|------|---------|----------|
| BERT | 110M | *bert-base-uncased* | English |
| DeBERTa | 100M | *microsoft/deberta-base* | English |
| XLNet | 110M | *xlnet-base-cased* | English |

Table 4: Details of pre-trained language models used in visualizations.

| | Original | Shuffled |
|---|---|---|
| Shuffled-3 | *An old man with a package poses in front of an advertisement .* | *An man old with package a poses in front of advertisement an .* |
| Shuffled-4 | A land rover is being driven across a river . | A land rover is being a driven river across . |
| Shuffled-5 | A man reads the paper in a bar with green lighting . | A man reads the paper in with green a lighting bar . |
| Shuffled-6 | A little boy in a gray and white striped sweater and tan pants is playing on a piece of playground equipment . | A little boy in striped a sweater and white gray and tan pants is playing piece playground of equipment on a . |
| Shuffled-SR | several women are playing volleyball . | volleyball are playing several women . |
| Shuffled-SR | a man and woman are sharing a hotdog . | a hotdog are sharing a man and woman . |

Table 5: Some cases of the shuffled SNLI datasets in our word swap probing. Texts in the same color mean the corresponding phrases.

by Wang et al. (2020), which feeds many repeated identical words to pre-trained language models and thus the attention values ($\alpha_{i,j}$ in Equation 5) are disentangled with contextual weights. More specifically, we randomly select 100 words from the corresponding vocabulary (filtering out single characters and sub-words such as "##nd"). We repeat each word to compose a sentence of length 128. These 100 sentences are fed into a language model and the attention weights across different layers are averaged as the positional weight matrix of a particular language model.

## B.2 WORD SWAP PROBING

To valid if language models with positional encodings are sensitive to the local and global word swaps, we construct Shuffle-$x$ and Shuffle-SR SNLI datasets. Shuffle-$x$ means the word orders of phrases with length $x$ are disrupted, e.g. "*an electric guitar*" is a 3-gram phrase, and it might be "*guitar an electric*" in Shuffle-3 SNLI. Through this way, a new sentence with the same meaning can be obtained and therefore the initial label of the sample will not be changed. To construct such shuffled datasets, the premise sentences in SNLI test set are shuffled and we keep the hypothesis sentences intact. Here, we let $x \in [3, 6]$ and select a subset from SNLI to make sure that every premise sentence has at least one phrase with length from 2 to 6. We select five types of target phrases for shuffling: *Noun Phrase, Verb Phrase, Prepositional Phrase, Adverb Phrase, and Adjective Phrase*. Finally, a Shuffle-$x$ SNLI is obtained by disrupting the order inside a phrase with length $x$ and the size for each shuffle-$x$ is around 5000. The first fourth rows in Table 5 shows some samples.

As for the Shuffle-SR SNLI dataset, the semantic roles of agent and patient are swapped in a sentence. We use the Algorithm 1 to collect a subset from SNLI test set. This algorithm is applied successively to the premise and hypothesis sentence for a sample whose label is entailment, and if the result of either of them is not null, we consider it a valid shuffled sample, which means we only shuffle the premise or hypothesis. After, we can obtain a new sample and the pair of sentences are contradicted with each other. In total, there are 1329 samples. To ensure that all sentences are semantically correct, we manually selected 300 pairs from them. The last two rows in Table 5 shows two examples in Shuffle-SR dataset.

To probe the capabilities of language models on our newly constructed datasets, we adopt five different pre-trained language models (as shown in Table 6) and we use Hugging Face for implementation (Wolf et al., 2020). These models are fine-tuned on the training set of SNLI, and the model with the best score on validation set is stored for the follow experiments. Note that there are off-the-

---

**Algorithm 1:** Construction of Shuffle-SR Sentences

**Input:**
$s$: a premise or hypothesis sentence in SNLI,
$\mathcal{A}$: Auxiliary verb list
$\mathcal{M}$: Semantic Role Labeling Model,
$\mathcal{D}$: Subject and Object Case Mapping // e.g., I ↔ me
**Output:** A sentence s* with shuffled agent and patient or *None*

1   $\mathcal{R} \leftarrow$ Predict the semantic roles of words in sentence $s$ by using the model $\mathcal{M}$
2   $\mathcal{V} \leftarrow$ Take the verb list from $\mathcal{R}$
3   **foreach** *verb $v$ in $\mathcal{V}$* **do**
4     **if** *$v$ appears in $\mathcal{A}$* **then**
5       continue
6     **if** *$\mathcal{R}$ does not contain an agent or patient* **then**
7       continue
8     $a, p \leftarrow$ Take the agent and patient from $\mathcal{R}$
9     s* $\leftarrow$ Swap the $a, p$ in sentence $s$
10     s* $\leftarrow$ Transform the subject and object case in s* if $a$ or $p$ in $\mathcal{D}$
11     **return** s*
12 **return** *None*

---

shell ALBERT and XLNet for natural language inference, we therefore use them directly without fine-tuning. During fine-tuning stage, the maximum length of the tokenized input sentence pair is 128, and the optimizer is Adam (Kingma & Ba, 2014) with learning rate of 2e-5. The batch size is 32 and the epoch is 3. After fine-tuning, the best model is evaluated on our shuffle SNLI test set, an we record their performances when faced with local and global word swaps.

### B.3   LINGUISTIC DISCUSSIONS OF LOCALITY AND SYMMETRY

*Locality* means that the positional weights favor the combination of units in a sentence to their adjacent units when creating higher-level representations. For example, sub-tokens can be composed into lexical meanings (e.g., {*"context"*, *"##ual"*} → *"contextual"*) or words can be composed into phrase-level meaning (e.g., {*"take"*, *"off"*} → *"take off"*), and clause-level and sentence-level meaning can be obtained through an iterative combination of low-level meanings, which is consistent with the multi-layer structure in pre-trained language models. From a linguistic perspective, words linked in a syntactic dependency should be close in linear order, which forms what can be called a dependency locality (Futrell et al., 2020). Dependency locality provides a potential explanation for the formal features of natural language word order. Consider the two sentences *"John throws out the trash"* and *"John throws the trash out"*. Both are grammatically correct. There is a dependency relationship between *"throws"* and *"out"* and the verb is modified by the adverb. However, language users prefer the expression with the first sentence because it has a shorter total dependency length (Dyer, 2017; Liu et al., 2017; Temperley & Gildea, 2018). Based on the visualizations and dependency locality, we, therefore, speculate that one main function that positional encodings have learned during pre-training is local composition, which exists naturally in our understanding of sentences. Empirical studies also demonstrate that performances of shuffled language models are correlated with the violation of local structure (Khandelwal et al., 2018; Clouatre et al., 2022).

The *symmetry* (also observed by Wang & Chen (2020); Wang et al. (2020)) of the positional matrices implies that the contributions of forward and backward sequences are equal when combining adjacent units under the locality constraint. This is contrary to our intuition, as the forward and backward tokens play different roles in the grammar, as we have seen in the examples of "*a man playing an electric guitar on stage*" and "*an electric guitar playing a man on stage*". However, this symmetry is less disruptive at the local level inside sentences. Recent work in psycholinguistics has shown that sentence processing mechanisms are well designed for coping with word swaps (Ferreira et al., 2002; Levy, 2008; Gibson et al., 2013; Traxler, 2014). Further, Mollica et al. (2020) hypothesizes that the composition process is robust to local word violations. Consider the following example:

    a. *on their last day they were overwhelmed by farewell messages and gifts*

    b. *on their last day they were overwhelmed by farewell and messages gifts*

    c. *on their last they day were overwhelmed farewell messages by and gifts*

The local word swaps (colored underlined words) are introduced in the latter two sentences, leading to a less syntactically well-formed structure. However, experimental results show that the neural response (fMRI blood oxygen level-dependent) in the language region does not decrease when dealing with word order degradation (Mollica et al., 2020), suggesting that human sentence understanding is robust to local word swaps. Likewise, symmetry can be understood in this way: when a reader processes a word in a sentence, the forward and backward nearby words are the most combinable, and the comprehension of this composition is robust to its inside order. On the other hand, symmetry is not an ideal property for sentence representations (consider the case of "*an electricity guitar*"), and we show the flaws of symmetry in the word swap probing task in Section 3.4.

### B.4 Details of Positional Encoders

To probe the correlations between the two properties and downstream tasks. We introduce a fully position-based encoder, which is adapted from the self-attention encoder. The key difference is that the attentional weights in the encoder are our handcrafted attenuated encodings, therefore, the locality and symmetry can be adjusted easily and we can observe the correlations caused by the changes of the two properties. An implementation example of this positional encoder is shown in Listing 2.

In this experiment, two sentence-level datasets, MR (Pang & Lee, 2005) and SUBJ (Pang & Lee, 2004), are used for evaluation. we use a single-layer and single-head positional attention as the encoder and the handcrafted encodings are fixed during training. We use the 840B-300d GloVe Pennington et al. (2014) vectors as word embeddings. For training, we use an Adam optimizer with an initial learning rate 0.002, and we use a decaying strategy to decrease the learning rate. We adopt a dropout method after the encoder layer, and train models to minimize the cross-entropy with a dropout rate of $0.5$. We train 5 epochs for each model and select the best model on dev sets to evaluate on test set. We repeat this procedure 5 times and use the average score to report.

### B.5 Details of Pre-training

We use the configuration of the original BERT$_{base}$ (Devlin et al., 2019) with 110M parameters for pre-training. Our model is implemented with PyTorch using the pytorchic-bert tool[2]. The number of layers, attention heads, and the projection dimension are 12, 12, and 768 respectively. We use the original vocabulary with a size of 30522. The training corpus is the English Wikipedia (20200101 dumps), which totals 13G after preprocessing by WikiExtractor. We pre-train with sequences of at most $T = 512$ tokens, and set the batch size as 64 to pre-training 600K steps. The optimizer is Adam with a learning rate of 5e-4, $\beta1 = 0.9$, $\beta2 = 0.999$, L2 weight decay of 0.01, and a warmup rate of 0.1. The dropout probability is always set as 0.1.

We use the original BERT$_{base}$ as our backbone and vary the positional encodings to pre-training different variants for comparison.

Listing 1 shows a code example about how to inject handcrafted positional encodings to the BERT backbone. Each variant is fine-tuned on the training dataset with different learning rates (among 9e-5, 7e-5, 5e-5, 3e-5, 1e-5). After, we evaluate the fine-tuned model on the Dev set and report the average score of five learning rates. Apart from BERT, we introduce the TUPE model as another baseline. Specifically, we pre-train the following variants:

- BERT is the original one and we use it as a baseline.

- BERT-$A^*$ and BERT-$I^*$ are varaints of the former two, but the encodings are learnable during pre-training.

- BERT-$A^*$-$s$ shares learnable positional encodings within a layer.

---

[2]`https://github.com/dhlee347/pytorchic-bert`

| Model | Size | Version | Fine-tuned by us |
|-------|------|---------|------------------|
| BERT | 110M | *bert-base-uncased* | ✓ |
| ALBERT | 223M | *ynie/albert-xxlarge-v2-snli_mnli_fever_anli_R1_R2_R3nli* | ✗ |
| DeBERTa | 100M | *microsoft/deberta-base* | ✓ |
| XLNet | 340M | *ynie/xlnet-large-cased-snli_mnli_fever_anli_R1_R2_R3-nli* | ✗ |
| StrucBERT | 340M | *bayartsogt/structbert-large* | ✓ |

Table 6: Details of pre-trained language models used in word swap probing.

- BERT-*only-c* is for ablation study, and the positional encodings $\mathbf{p}_i$ in Equation 2 are removed.

- BERT-*only-p* is for ablation study, and the contextual encodings $\mathbf{x}_i$ in Equation 2 are removed.

- BERT-$A^*$-*Seq* combines the two features in a sequential way, and the positional attentions are first used and then contextual attentions.

- ALiBi adds linear biases to contextual weights proposed by Press et al. (2021a), and we apply a softmax layer to the original biases for obtaining a attention weight vector.

- ALiBi-*Seq* uses the same biases with ALiBi but combines the two features in a sequential way.

Suppose that the hidden dimension is 768, the layer number is 12, the head number is 12, and the maximum length is 512 for BERT$_{base}$ model, we can calculate the size for each variant. The number of parameters of handcrafted positional encoding for each head is 262K ($512 \times 512$). If positional heads are different across all layers, the total cost is 37.7M ($512 \times 512 \times 12 \times 12$). If the positional encodings are shared across heads, the total cost is 3.1M ($512 \times 512 \times 12$).

```python
class MultiHeadedSelfAttention(nn.Module):
    """ Multi-Headed Scaled Dot Product Attention """
    def __init__(self, config):
        super().__init__()
        self.n_heads = config.n_heads
        self.drop = nn.Dropout(config.p_drop_attn)
        self.proj_q = nn.Linear(config.dim, config.dim)
        self.proj_k = nn.Linear(config.dim, config.dim)
        self.proj_v = nn.Linear(config.dim, config.dim)

    def forward(self, x, mask, pe):
        """
        x, q(query), k(key), v(value) : (B(batch_size), S(seq_len), D(dim))
        mask : (B(batch_size) x S(seq_len))
        pe: positional weights (B(batch_size), H(Head_number)), S(seq_len), S(seq_len))
        * split D(dim) into (H(n_heads), W(width of head)) ; D = H * W
        """
        # (B, S, D) -proj-> (B, S, D) -split-> (B, S, H, W) -trans-> (B, H, S, W)
        q, k, v = self.proj_q(x), self.proj_k(x), self.proj_v(x)
        q, k, v = (split_last(x, (self.n_heads, -1)).transpose(1, 2)
                   for x in [q, k, v])
        # (B, H, S, W) @ (B, H, W, S) -> (B, H, S, S) -softmax-> (B, H, S, S)
        scores = q @ k.transpose(-2, -1) / np.sqrt(k.size(-1))

        # inject positional weights into contextual weights
        # (B, H, S, S) + (B, H, S, S) -> (B, H, S, S)
        scores = scores + pe

        if mask is not None:
            mask = mask[:, None, None, :].float()
            scores -= 10000.0 * (1.0 - mask)

        scores = self.drop(F.softmax(scores, dim=-1))
        # (B, H, S, S) @ (B, H, S, W) -> (B, H, S, W) -trans-> (B, S, H, W)
        h = (scores @ v).transpose(1, 2).contiguous()
        # -merge-> (B, S, D)
        h = merge_last(h, 2)
        return h
```
Listing 1: A code example of how to inject handcrafted positional encodings into self-attentions.

To inject handcrafted positional encodings, we pre-compute the positional weights and add them to the contextual weights directly, as shown in Listing 1. These weights can be either frozen or learnable during pre-training. The code of the sequence combination is shown in Listing 3.

## B.6 DETAILS OF DOWNSTREAM DATASETS

SentEval is based on a set of existing text classification tasks involving one or two sentences as input. However, most tasks in SentEval are closely related to sentiment analysis and thus not diverse enough. GLUE benchmark introduces a series of difficult natural language understanding tasks while some particular tasks only contain one dataset, e.g., sentiment analysis and textual similarity. Moreover, the size of WNLI in GLUE is rather small and the GLUE webpage notes that there are issues with the construction of this dataset [3]. To better evaluate the capability of models for sentence representation, we, therefore, select 10 datasets from SentEval and GLUE, covering four types of sentence-level tasks:

- **Sentiment Analysis** is also known as opinion mining, which aims to classify the polarity of a given text, whether the expressed opinion is positive, negative, or neutral. We use MR (Pang & Lee, 2005), SUBJ (Pang & Lee, 2004), and SST (Socher et al., 2013) for this task.

- **Textual Entailment** describes the inference relation between a pair of sentences, whether the premise sentence entails the hypothesis sentence. Actually, this is a classification task with three labels: entailment, contradiction, and neutral. Here, we use QNLI (Rajpurkar et al., 2016), RTE (Dagan et al., 2005; Haim et al., 2006; Giampiccolo et al., 2007; Bentivogli et al., 2009) and MNLI (Williams et al., 2018) for evaluation. Note that we report the average score for the two test sets of MNLI.

- **Paraphrase Identification** is to determine whether a pair of sentences have the same meaning. We use MRPC (Dolan & Brockett, 2005) and QQP [4] for evaluation.

- **Textual Similarity** deals with determining how similar two pieces of texts are. We use STS-B (Cer et al., 2017) and SICK-R (Marelli et al., 2014) for evaluation.

```python
class MultiHeadPositionalAttention(nn.Module):
    """ Multi-Headed Scaled Dot Product Attention """
    def __init__(self, config):
        super().__init__()
        self.n_heads = config.n_heads
        self.drop = nn.Dropout(config.p_drop_attn)

    def forward(self, x, mask, pe):
        """
        x, q(query), k(key), v(value) : (B(batch_size), S(seq_len), D(dim)
        mask : (B(batch_size) x S(seq_len))
        pe: positional weights (B(batch_size), H(Head_number)), S(seq_len), S(seq_len))
        * split D(dim) into (H(n_heads), W(width of head)) ; D = H * W
        """
        # (B, S, D) -proj-> (B, S, D) -split-> (B, S, H, W) -trans-> (B, H, S, W)
        q, k, v = (split_last(x, (self.n_heads, -1)).transpose(1, 2)
        for x in [q, k, v])
        # (B, H, S, W) @ (B, H, W, S) -> (B, H, S, S) -softmax-> (B, H, S, S)

        scores = pe
        if mask is not None:
            scores.masked_fill_(~mask, 0.)

        # (B, H, S, S) @ (B, H, S, W) -> (B, H, S, W) -trans-> (B, S, H, W)
        h = (scores @ v).transpose(1, 2).contiguous()
        # -merge-> (B, S, D)
        h = merge_last(h, 2)
        return h
```

Listing 2: A code example of the Positional Attention.

---

[3] https://gluebenchmark.com/faq
[4] data.quora.com/First-Quora-Dataset-Release-Question-Pairs

# C  ADDITIONAL EXPERIMENTS

## C.1  LOSS CURVES OF PRE-TRAINING

Apart from the performances on downstream tasks, the loss curves are also checked for different variants. For this goal, the training loss and validation loss are stored after certain steps. We use a hold-set as the validation set. As shown in Figure 6, our proposed BERT-A$^*$ and BERT-A$^*$-*Seq* have smaller loss than the original BERT. This can be observed again on the validation set.

## C.2  ABLATION STUDY OF POSITIONAL AND CONTEXTUAL ENCODINGS

To check the importance of positional and contextual Encodings, we conduct an ablation study. For this goal, the contextual encodings $\mathbf{x}_i$ or positional encodings $\mathbf{p}_i$ in Equation 2 are removed, respectively, during pre-training and the two new models are evaluated on 10 sentence-level datasets. As shown in Table 7, the BERT-*only-c* and BERT-*only-p* both lag behind the original BERT models, which means the combination of the two features is beneficial for sentence representations. On the other hand, positional encodings are more important for sentiment analysis, and the cross-attentions from contextual embeddings matter in sentence-pair tasks.

| Model | Sentiment Analysis | | | Textual Entailment | | | Paraphrase Identification | | Textual Similarity | | |
|---|---|---|---|---|---|---|---|---|---|---|---|
| | MR (22K) | SUBJ (20K) | SST-2 (68.8K) | QNLI (110K) | RTE (5.5K) | MNLI (413K) | MRPC (5.4K) | QQP (755k) | STS-B (8.4K) | SICK-R (9.4K) | Avg |
| BERT | $72.5_{\pm5.3}$ | $91.0_{\pm2.7}$ | $86.4_{\pm2.7}$ | $85.8_{\pm1.0}$ | $59.2_{\pm1.2}$ | $78.2_{\pm0.8}$ | $73.5_{\pm1.8}$ | $88.7_{\pm0.6}$ | $77.8_{\pm4.1}$ | $64.9_{\pm6.0}$ | 77.8 |
| BERT-*only-c* | $73.0_{\pm4.6}$ | $88.9_{\pm2.5}$ | $82.9_{\pm0.4}$ | $82.0_{\pm0.1}$ | $62.7_{\pm4.3}$ | $70.8_{\pm0.8}$ | $74.1_{\pm0.3}$ | $86.9_{\pm0.5}$ | $78.5_{\pm0.3}$ | $64.5_{\pm5.7}$ | 76.2 |
| BERT-*only-p* | $73.8_{\pm4.6}$ | $90.8_{\pm1.4}$ | $84.0_{\pm0.7}$ | $79.8_{\pm1.2}$ | $50.9_{\pm1.4}$ | $68.3_{\pm1.0}$ | $73.9_{\pm1.6}$ | $85.8_{\pm0.6}$ | $47.1_{\pm18.0}$ | $51.7_{\pm9.2}$ | 70.6 |

Table 7: Ablation study across 10 sentence-level tasks. We report the average score of five runs using different learning rates.

```python
class Sequence(nn.Module):
    """ Sequence Block """

    def __init__(self, config):
        super().__init__()
        self.pos_mode = config.pos_mode
        self.pos_learnable = config.pos_learnable
        self.self_attention = MultiHeadedSelfAttention(config)
        self.positional_attention = PositionalAttention(config, learnable=self.pos_learnable)

    def forward(self, x, mask):
        # positional attention
        pa = self.positional_attention(x, mask)
        # contextual attention
        sa = self.self_attention(pa, mask)
        return sa
```

Listing 3: A code example of the Sequence combination of positional and contextual features.

## C.3  RESULTS OF LINGUISTIC PROBING

To understand the linguistic role of positional encodings, we adopt the probing tasks proposed by Conneau et al. (2018), which feeds sentence representations obtained by neural models to a series of linguistic classification tasks. The sentences for all our tasks are extracted from the Toronto Book Corpus (Zhu et al., 2015), pre-processed by Paperno et al. (2016) and parsed by using the pre-trained PCFG model (Klein & Manning, 2003). We use the standard SentEval (Conneau & Kiela, 2018) to test every model. For obtaining sentence representations, we apply the *mean* pooling strategy to the output of each layer of a frozen language model, which is suggested by Reimers & Gurevych (2019). We follow the standard setup and run a grid search for selecting the layer with the best score on the dev set, and report the best score across layers for each probing task (Table 9), which represents the linguistic ability of each model. The following are the details of each probing task, including three categories:

- SentLen (Surface) aims to predict the length of sentences in terms of the number of words, and the dataset is constructed following Adi et al. (2017).

| Relation | Distance | *contextual* | *positional* | *both* |
|---|---|---|---|---|
| Root | 0.0 | 99.3 | 3.8 | 86.5 |
| auxpass | 1.1 | 44.6 | 91.1 | 92.9 |
| compound | 1.2 | 21.7 | 75.0 | 70.6 |
| aux | 1.3 | 25.2 | 77.9 | 79.1 |
| nummod | 1.3 | 26.8 | 78.9 | 75.5 |
| amod | 1.4 | 19.7 | 69.3 | 66.9 |
| det | 1.8 | 47.9 | 52.9 | 51.6 |
| advmod | 2.1 | 16.5 | 62.4 | 58.7 |
| pobj | 2.3 | 9.0 | 33.9 | 46.3 |
| nsubj | 2.3 | 13.4 | 58.2 | 52.6 |
| poss | 2.3 | 15.9 | 31.7 | 43.5 |
| dobj | 2.5 | 20.0 | 34.8 | 41.6 |
| prep | 2.6 | 23.1 | 62.8 | 63.4 |
| npadvmod | 3.1 | 14.4 | 30.0 | 43.8 |
| cc | 3.1 | 28.4 | 52.0 | 51.6 |
| mark | 4.1 | 25.1 | 21.3 | 64.4 |
| conj | 4.9 | 25.1 | 31.2 | 33.6 |
| punct | 7.3 | 25.9 | 30.3 | 32.7 |
| advcl | 7.6 | 18.4 | 9.8 | 20.1 |
| ccomp | 8.1 | 29.0 | 12.4 | 33.2 |
| short | $\leq 4$ | 28.4 | 54.3 | 61.6 |
| long | $> 4$ | 24.7 | 21.0 | 36.8 |
| Macro Avg | - | 27.5 | 46.0 | 55.4 |

Table 8: Evaluations of predictions of dependency relations on MRPC dataset. The top-20 common relations are shown. The distinction of *"short"* and *"long"* is whether the average length of the relation is greater than 4.

- WC (Surface) means word content, which checks whether it is possible to recover information about the original word from the embedding of the sentence.

- BShift (Syntactic) means bigram shift. In this task, two random adjacent words in a sentence are swapped and the goal is to detect if a model is sensitive to legal word orders.

- TreeDepth (Syntactic) tests whether a model can infer the depth of the syntactic tree of sentences.

- TopConst (Syntactic) tests whether a model can recognize the top constituents of the sentence, e.g., "*[Then] [very dark gray letters on a black screen] [appeared] [.]*" has top constituent sequence: "ADVP NP VP ". This dataset is first introduced by Shi et al. (2016).

- Tense (Semantic) asks for the tense of the main clause verb.

- SubjNum (Semantic) focuses on the number of the subject of the main clause.

- ObjNum (Semantic) tests for the number of the direct object of the main clause.

- SOMO (Semantic) checks the sensitivity of a model to random replacement of a noun or verb.

- CoordInv (Semantic) tests whether a model can recognize the order of clauses is inverted.

## C.4 DEPENDENCY ANALYSIS OF POSITIONAL ENCODINGS

The detailed scores of each relation are shown in Table 8. We find that positional attentional heads outperform contextual heads on short-distance relations, e.g., `auxpass` and `compound`. Contextual attention can capture better long-distance relations than positional attention while contextual attention itself has a certain degree of ability to detect some long-distance relations such as `conj` and `punct`. Note that there exists a head in contextual attention maps attending the token itself, therefore, the score on the `Root` relation is the best.

## C.5 VISUALIZATIONS

In Figure 1, we visualize the averaged positional weights of various pre-trained language models and identify they have similar visualized results. However, We find that the behavior of positional encodings is very diverse across attention heads. Note that there are 144 attentional heads (12 layers ×

| Model | Surface | | Syntactic | | | Semantic | | | | |
|-------|---------|------|-----------|----------|--------|-------|---------|--------|------|---------|
|       | SentLen | WC   | TreeDepth | TopConst | BShift | Tense | SubjNum | ObjNum | SOMO | CoordInv |
| BERT   | **92.55** | 23.54 | **38.75** | **77.21** | **77.67** | **85.30** | **84.50** | **83.94** | **52.43** | **61.22** |
| BERT-*c* | 89.22 | **57.59** | 32.86 | 56.65 | 51.16 | 82.94 | 78.97 | 81.63 | 51.86 | 50.35 |
| BERT-*p* | 89.64 | 38.27 | 33.78 | 63.15 | 53.68 | 67.17 | 71.17 | 66.73 | 51.19 | 57.42 |
| TUPE   | 90.08 | 85.76 | **39.14** | **78.38** | **79.44** | **87.13** | **86.50** | **86.36** | **53.78** | **62.60** |
| TUPE-*c* | **91.85** | 62.42 | 34.21 | 58.63 | 50.86 | 83.43 | 80.63 | 81.96 | 51.75 | 50.74 |
| TUPE-*p* | 90.07 | 65.59 | 37.74 | 77.58 | 75.32 | 77.67 | 82.13 | 80.40 | 51.96 | 62.07 |

Table 9: Linguistic role probing of positional encodings. *c* and *p* means only using contextual and positional weights, respectively.

12 heads) for the BERT$_{base}$ model. For example, the visualizations of BERT (Figure 8) validate this phenomenon. Besides, we observe that BERT exhibits a hierarchical structure: positional weights of lower layers are nearly uniform (Layer-4), middle layers attend more to local units (Layer-7) and higher layers demonstrate the asymmetric property (Layer-12). We also visualize all the positional heads in BERT-A$^*$ (Figure 9) and BERT-A$^*$-*Seq* (Figure 10).

**Visualization Analysis of BERT-A$^*$.** BERT-A$^*$ outperforms BERT by 3.1 percentage points on average across 10 downstream tasks. The main difference between BERT-A$^*$ and BERT is the learnable handcrafted positional encodings. For visualizations, we take the positional weight $\delta_{i,j}$ in Equation 5 instead of Identical Word Probing.. Figure 9 shows that most positional heads perfectly satisfy the properties of locality and symmetry, which can bring better inductive bias for sentence representations. Another observation is that the diagonal bandwidths are diverse across positional heads after learning, which means proximity units can be combined at different distances. We conclude that, Compared to randomly initialized positional encodings, the encodings initialized with locality and symmetry properties lead to better sentence representation models.

**Visualization Analysis of BERT-A$^*$-*Seq*.** Compared to BERT-A$^*$, BERT-A$^*$-*Seq* can further improve the scores on downstream tasks by combining positional and contextual features in a sequential way. Figure 10 visualize all positional heads in BERT-A$^*$-*Seq*, we observe these matrices are still perfectly localized and symmetrical, which is similar to the behaviors of BERT-A$^*$. We conclude that the Sequence combination can yield better sentence representations than the traditional Add combination.

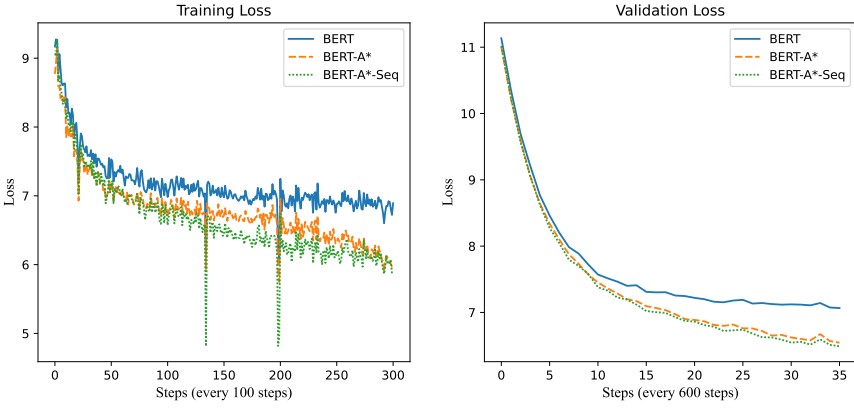

Figure 6: Loss curves of different variants adapted from BERT$_{base}$ model.

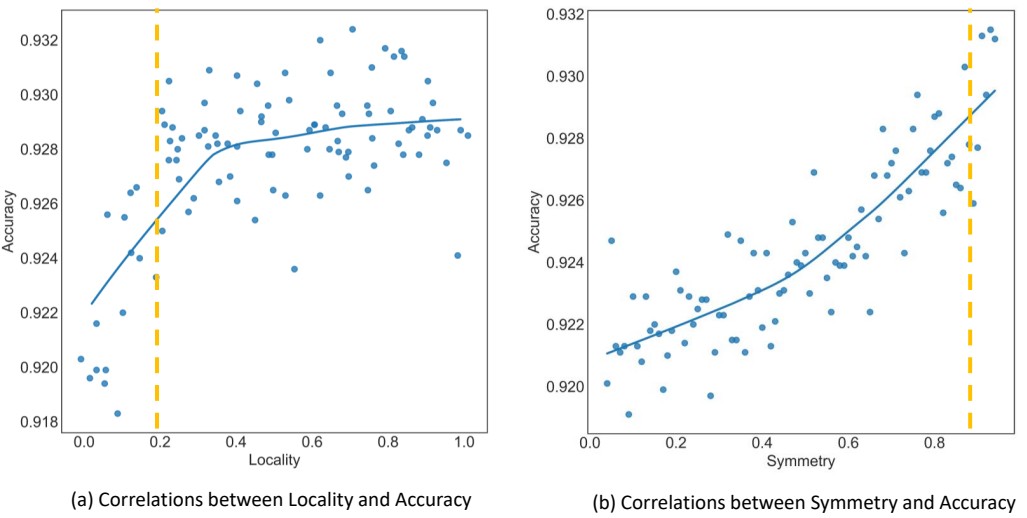

(a) Correlations between Locality and Accuracy

(b) Correlations between Symmetry and Accuracy

Figure 7: Correlations between the two properties (Locality and Symmetry) and accuracy on SUBJ dataset (Pang & Lee, 2004). The yellow line shows the locality or symmetry of the pre-trained BERT.

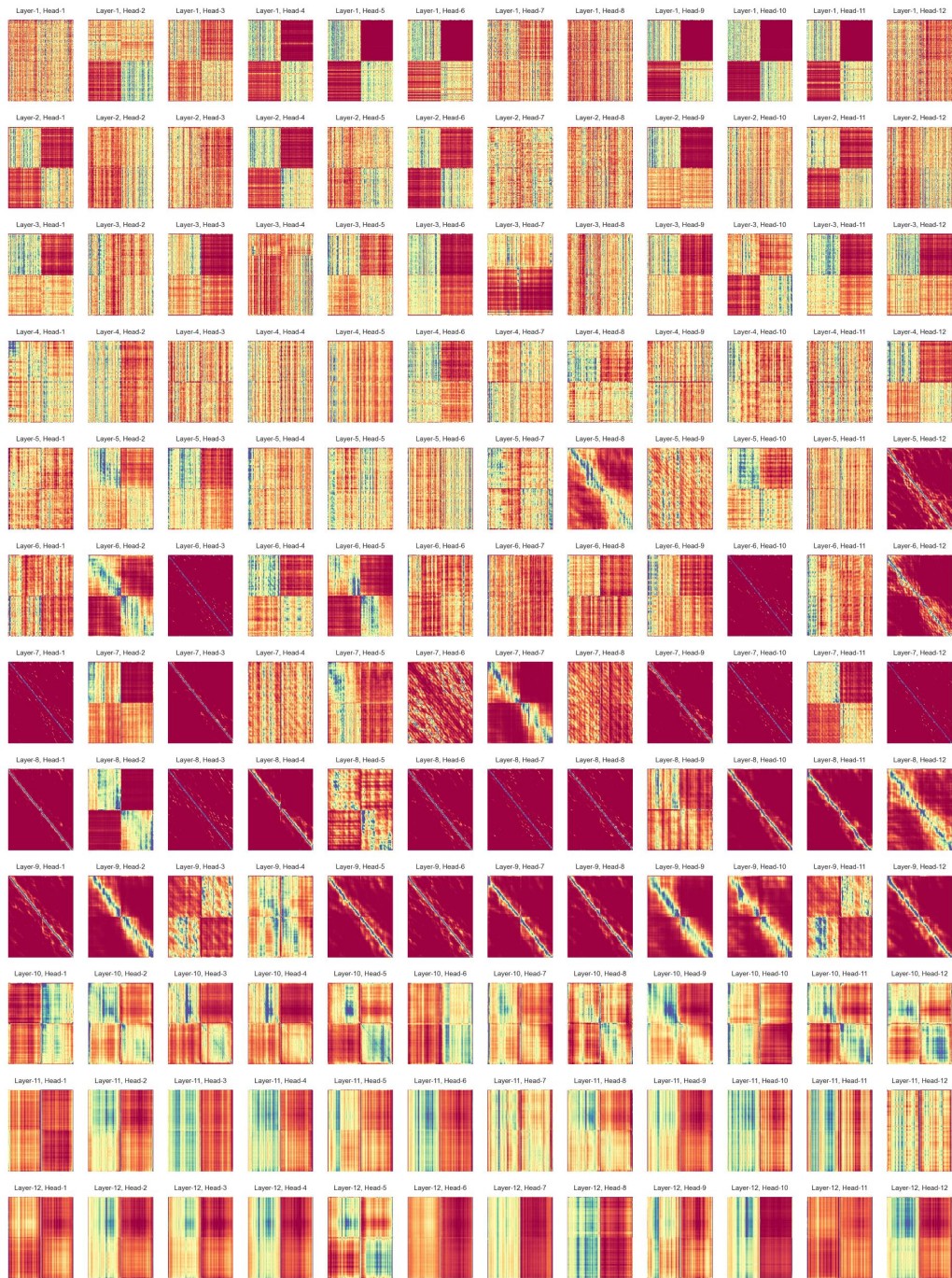

Figure 8: Visualizations of positional weights of BERT across all layers. The weights are computed by Identical Word Probing. Red color means lower values and blue color means higher values.

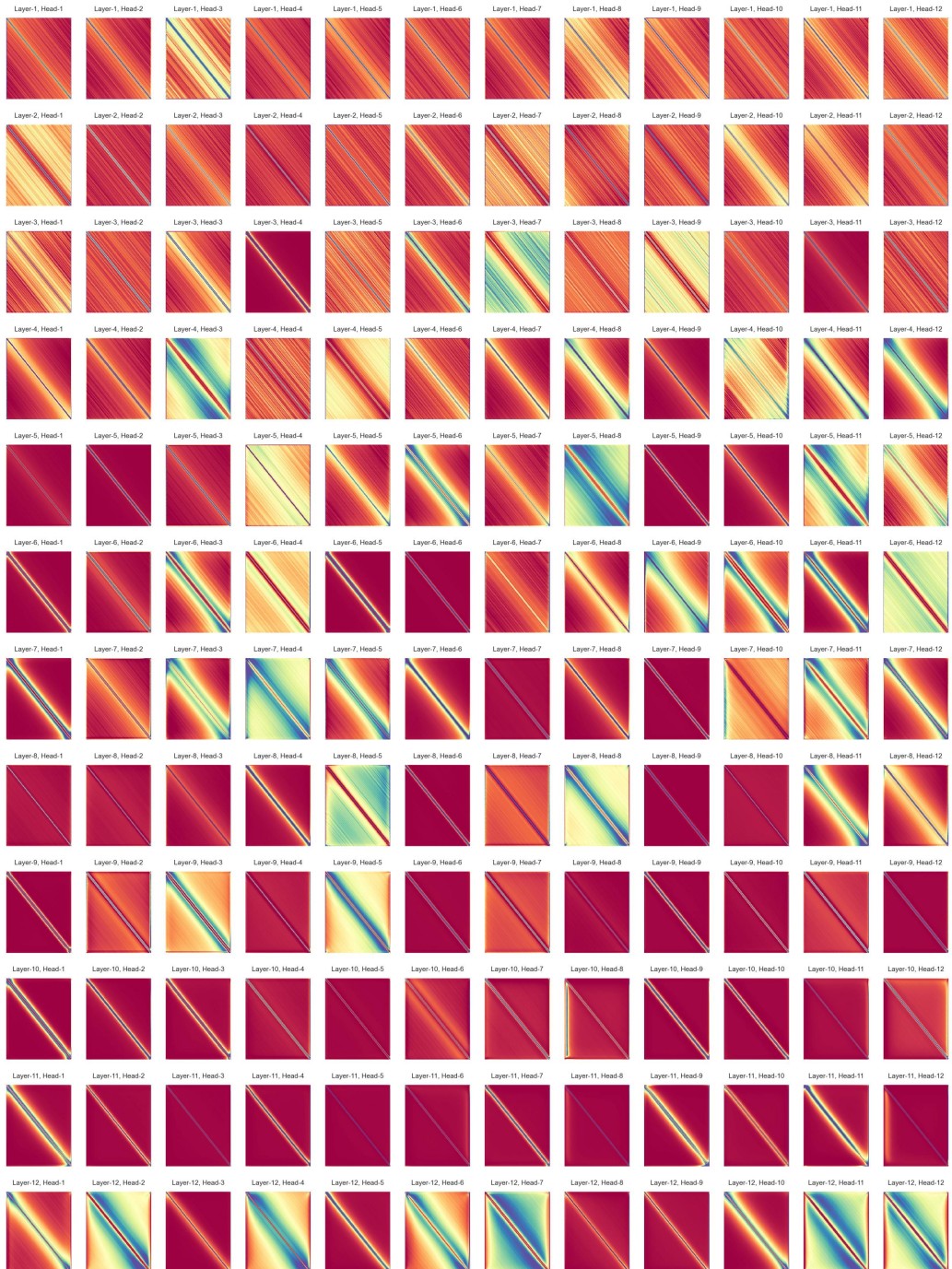

Figure 9: Visualizations of positional weights of BERT-A* across all layers. Red color means lower values and blue color means higher values.

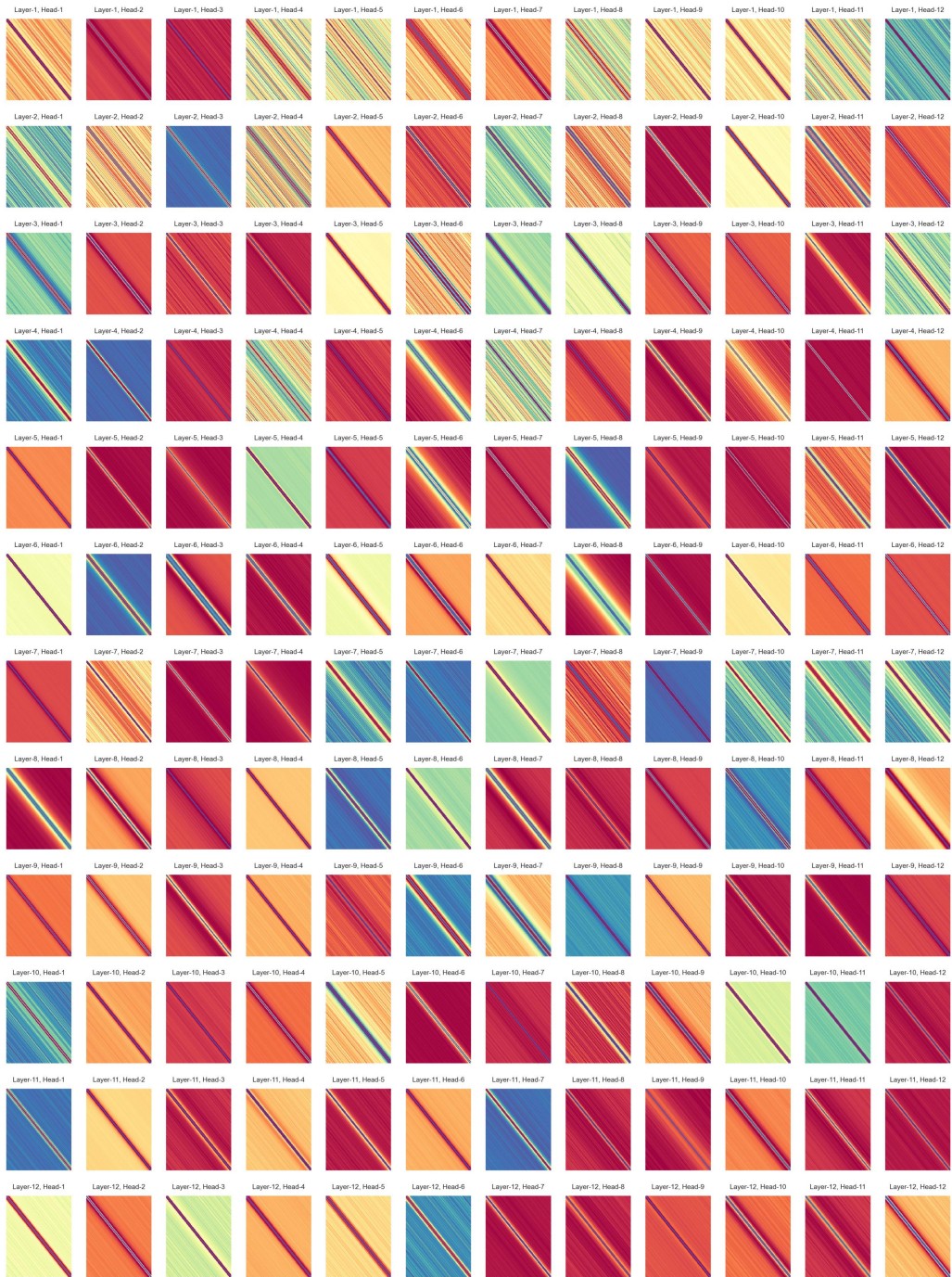

Figure 10: Visualizations of positional weights of BERT-A*-*Seq* across all layers. Red color means lower values and blue color means higher values.

