# OpenReview forum: "UNDERSTANDING THE ROLE OF POSITIONAL ENCODINGS IN SENTENCE REPRESENTATIONS"
_ICLR.cc/2023/Conference — Submitted to ICLR 2023_

### Official Review · Reviewer_gutD · 2022-10-21

**Confidence:** 4
**Correctness:** 2
**Technical Novelty And Significance:** 2
**Empirical Novelty And Significance:** 1
**Recommendation:** 5

**Clarity, Quality, Novelty And Reproducibility:**

There are some significant details left out about the sequence combination. The paper needs some extensive proofreading for grammar/typos/local errors, but is otherwise clear and well structured. The novelty is limited, as the existing literature contains both behavioral analysis of word swaps and simplified alternatives to position embeddings based on similar intuitions. The quality of these experiments is reasonable, but I'm not convinced that they offer strong support for the authors' intuitions.

**Strength And Weaknesses:**


Strengths:
- I find the question they are attempting to ask about the role of position encodings to be an interesting one.
- I have never seen a clear discussion of the symmetry property, although it is evident in attention heat maps in every paper on transformers.
- More importantly, I have never seen a discussion of the potential drawbacks of a symmetry bias. It’s not something that I had considered before, and it makes me curious about human cognition: are we better able to handle symmetric word swaps than random word swaps, for example? I expect not. This is a clear example of cognitively implausible behavior at the position encoding level, although to be fair nobody is claiming any kind of cognitive plausibility in position encoding. I would like to see the focus on symmetry expanded.
- I was impressed by the incorporation of research in psycholinguistics to discuss word swaps. I would actually have liked to see a longer discussion of the implications of the different studies, in terms of local and more distant word swaps.
- In future work or versions of this paper, I would be extremely interested to see the symmetry and locality biases treated separately.
- The new method of Sequence Combination as an alternative to adding together position and contextual encodings appears to work well.

Weaknesses:
- My main problem with this paper is that I do not find the experiments to be particularly convincing evidence of the properties that they discuss. In particular, they use a new encoding scheme to artificially impose strict symmetry and locality, and find that performance is somewhat damaged but not too much. I don’t know what result they would consider to be evidence against the importance of symmetry and locality, perhaps some very extreme blow to performance? In general, I don’t think that the performance of these simplistic position encodings give any particular evidence as to whether their biases are *also* the reason why the original encoding scheme worked.
- I would have liked to see more variety in the position encoding schemes being used, rather than using only BERT’s default position encoding.
- When you set the contextual or position encoding to zero, you create a significant distribution shift in representations. Perhaps taking an average would be better? In general, I’m skeptical about these kinds of causal interventions on a fully trained model. You are changing the expected norm and representational geometry, and that might change to different degrees depending on which component you are removing.
- They seem to be arguing that symmetry in the attention weights means that you can swap the position of different tokens without changing the meaning of the sentence, but that’s not clear to me? I feel like I need an explanation that either gives the mathematical process or diagrams the reason why this would be the effect.
- I would like to see for learned position encodings whether symmetry is a learned property.
- The experiments on simplistic position encodings bear a significant resemblance to Alibi, and I’m not sure what they contribute over that scheme in terms of understanding https://arxiv.org/pdf/2108.12409.pdf
- There are nonlinear interactions between the position encodings and the context. Currently they consider these interactions to be part of the position encoding, but I’m not convinced that that’s appropriate. It’s possible that some of the bias they are talking about is due to the isolated role of position $\frac{(p_iW^Q)(p_jW^K)^T}{\sqrt{d}}$ so I think that experiments should separately consider what happens when interactions are included or removed.
    - When you measure the effect of removal, do you just straightforwardly remove it or do you take an average value or do you retrain the network to work without the encoding in question? There is a big difference between these strategies because you are creating a very large domain shift by removing a component of the representation, but that doesn’t necessarily mean that the information of structure within that component is as important as it seems based on changes effected by the wholesale removal of a component.
    - Maybe you should be considering something like shapley values.
- There should be an explicit comparison between their word swap probe and other existing methods for behaviorally testing the role of word order. Currently I’m not clear on what the new contributions are of this particular experiment.
- Do you have a metric for measuring symmetry? That would be better than all of the figures. Even something simple like the matrix norm of the difference between the original attention matrix and its transpose.
- There is a lack of detail about the implementation of sequence combination, although it seems like it should be one of these strongest points in the paper.

Minor/references:
- This paper needs extensive editing for typos (“encoings”, mismatched subject-object number, etc).
- Probably should add a note as to why autoregressive models aren’t relevant to position encodings research http://arxiv.org/abs/2203.16634
- Sharp Nearby, Fuzzy Far Away: How Neural Language Models Use Context https://arxiv.org/pdf/1805.04623.pdf
- When you describe the symmetry property, make sure to clarify the axis of symmetry. Even with the example given, which initially just confused me, it took me a while to understand that this was referring to equidistant tokens having the same weight rather than the same weight applying both to both key-query and query-key directions.
- “BEET-A∗-s” I don’t understand what this string means.
- “we adopt widely used 10 probing tasks” — make sure that you actually cite each of the tasks separately, rather than only the benchmark paper that aggregates them.
- “For surface tasks, the surface knowledge is stored more in bottom layer …” I find this a confusing sentence
- Don’t start a sentence with a variable name if you can help it, as in the paragraph before equation 10 (https://jmlr.csail.mit.edu/reviewing-papers/knuth_mathematical_writing.pdf)




**Summary Of The Paper:**

This paper investigates the role of position encodings in contrast to contextual encodings in BERT-based models. They illustrate two biases of these encodings: locality (a tendency to focus on nearby tokens) and symmetry (a tendency to focus at a particular distance from the target to the same degree whether the distance is measured backwards or forwards). They decompose attention into the sum of position and contextual components, and ablate each component to measure the role of the remaining representation. They then propose several modifications to the encoding scheme in order to test their intuitions about the biases of the default scheme (one of these modifications, sequence combination, appears to lead to a performance improvement).

**Summary Of The Review:**

The paper has two potentially significant contributions: a discussion of the drawbacks of symmetry bias in position encoding and a new (to my understanding) method of "sequence combination" that presents an alternative to summation of position and contextual encodings.

However, the other experimental results and methods proposed are unconvincing to me as evidence of their claims of the importance of locality and symmetry. Furthermore, they are very similar to existing work in the analysis and architecture literature, most of which is uncited here.

---

> ### Author Response · Authors · 2022-11-16
> **Reply to Reviewer gutD (1/2)**
>
> Dear Reviewer gutD,
>
> **We appreciate the time you spent reviewing the manuscript and the constructive and insightful comments you proposed. We revised the manuscript and added additional experiments based on your suggestions. Let us now address your questions and concerns one by one.**
>
> > My main problem with this paper is that I do not find the experiments to be particularly convincing evidence of the properties that they discuss.
>
> **A1:** We have revised the manuscript and introduced two quantitative metrics: The *Locality* is a metric that depicts the degree of the gathering of weights in local positions for an attentional weight vector while the *Symmetry* is a metric that describes how symmetrical the weights scatter around the current position for an attentional weight vector (The formulas are in Section 3.1.)
>
> We use these quantitative metrics to answer two questions (Section 3.2).
> * Are these two properties learned during pre-training?
> * Do the two properties matter?
>
> To answer the first question, we compute the locality and symmetry for learned and fixed encodings. Experimental results show that encodings after pre-training become more local and symmetrical, which shows the two properties are indeed learned. As for fixed ones, they are intentionally designed to have better locality and symmetry.
>
> To answer the second question, we study the correlations between the two properties and the performances of downstream tasks. Empirical results demonstrate that better locality and symmetry can yield better sentence representations.
>
> > *"I don’t think that the performance of these simplistic position encodings give any particular evidence as to whether their biases are also the reason why the original encoding scheme worked."*
>
> In the handcrafted encoding probe, we initialize the language model with encodings that mimic the locality (0.17) and symmetry (1.0) of a pre-trained BERT. We observe that the new variant can significantly outperform the original BERT, which shows the two properties can bring better inductive bias and further explain the main function of positional encodings.
>
> > I would have liked to see more variety in the position encoding schemes being used, rather than using only BERT’s default position encoding.
>
> **A2:** We now included ALiBi in our experiments, and the results below show that the sequence combination can bring consistent improvements (see Table 3).
>  |  Model   | Score  |
> |  ----  | ----  |
> | A*  | 81.8 |
> | A*-seq  | 82.5 |
> | AliBi  | 79.7 |
> | AliBi-seq  | 80.4 |
>
> > When you set the contextual or position encoding to zero, you create a significant distribution shift in representations.
>
> **A3:** The shift in distribution may happen slightly but does not weaken our main conclusions. We only hide the contextual correlation $\gamma_{i,j}$ in Equation 5 of a pre-trained BERT (instead of removing the contextual encodings completely) and thus the attentional weight $\alpha_{i,j}$ depends only on the positional correlation. Note that this operation does not destroy the structure of the original network because a softmax layer is applied to the vector $\mathbf{\alpha_{i}}$, and the output is still an attentional weight vector that can be regarded as a kind of discrete probability distribution. Therefore, the output sentence representation is somewhat decoupled from contextual encoding.
> We clarified this in Section 4.1.
>
> > They seem to be arguing that symmetry in the attention weights means that you can swap the position of different tokens without changing the meaning of the sentence, but that’s not clear to me? I feel like I need an explanation that either gives the mathematical process or diagrams the reason why this would be the effect.
>
> **A4:** In our manuscript, we claim that *“this symmetry is less disruptive at the local level inside sentences.”*  and  *”it is worth mentioning that the random shuffling may shift the semantics of the original sentence and thus cause the change of labels. ”*
> However, prior probes assume that the random shuffling does not change the semantics, which might be problematic. This is the reason that we propose two new probing tasks. We will explain this in A8.
>
> > I would like to see for learned position encodings whether symmetry is a learned property.
>
> **A5:** See A1.

---

> > ### Author Response · Authors · 2022-11-16
> > **Reply to Reviewer gutD (2/2)**
> >
> > > The experiments on simplistic position encodings bear a significant resemblance to Alibi, and I’m not sure what they contribute over that scheme in terms of understanding
> >
> > **A6:** There are two key differences between our encodings and other manually designed ones such as the T5 bias and ALibi.
> > First, the output generated by our method is an attentional vector (or a discrete probability distribution) that can be regarded as a type of attention mechanism. Thus, we can estimate the locality and symmetry individually. ALiBi biases, in contrast, cannot be measured by our proposed metrics directly.
> > Second, we can adjust the hyper-parameters in our method for obtaining encodings with different localities and symmetry, which ALiBi does not allow.
> >
> > > There are nonlinear interactions between the position encodings and the context. Currently they consider these interactions to be part of the position encoding, but I’m not convinced that that’s appropriate.
> >
> > **A7:** TUPE  (Equation 3) has no content-position interactions anymore. Hence, the probing results are not entangled with content-position correlations and therefore are more reliable. BERT still contains two content-position interactions (Equation 2) but their impact on the output representations is negligible, as proved in the original TUPE paper.
> >
> > > There should be an explicit comparison between their word swap probe and other existing methods for behaviorally testing the role of word order. Currently I’m not clear on what the new contributions are of this particular experiment.
> >
> > **A8:** Existing probes study the sensitivity of language models to word order by shuffling sentences, and they can be roughly divided into three categories: random swap (pham2021out, gupta2021bert, abdou2022word), n-gram swap (sinha2021masked) and subword-level phrase swap (clouatre2022local).
> > It is worth mentioning that these methods of word swap may change the semantics of the original sentences and thus cause the change of labels.
> > However, they assume that the labels of the randomly shuffled sentences are unchanged and thus may introduce biases into the probing results.
> >
> > To address the issue, we propose two new probing tasks of word swaps: **Constituency Shuffling** and **Semantic Role Shuffling**.
> > Constituency Shuffling aims to disrupt the inside order of constituents, which is able to change the word order while preserving the maximum degree of original semantics.
> > Semantic Role Shuffling intentionally changes the semantics by swapping the semantic roles in a sentence
> >
> > Our new probing results find a potential weakness of existing PEs, which is ignored by prior studies.
> >
> > > Do you have a metric for measuring symmetry?
> >
> > **A9:** See A1
> >
> > > There is a lack of detail about the implementation of sequence combination, although it seems like it should be one of these strongest points in the paper.
> >
> > **A10:**
> > The old version described the implementations in the appendix. Page 20 gives the implementation of sequence combination (Listing 3). We now refer to this explicitly in Section 7.
> >
> > > When you describe the symmetry property, make sure to clarify the axis of symmetry.
> >
> > **A11:** We added more details on this in Section 3.1
> >
> > > “BEET-A∗-s” I don’t understand what this string means.
> >
> > **A12:** This model uses learned encodings (symbol *) but the encodings are shared in a particular layer (letter s), i.e., all attentional heads in a layer have the same encoding. We have clarified this in the revised manuscript (Section 3.3)
> > We use this model to show that BERT obtains better inductive bias if it is initialized with encodings that have good locality and symmetry. At the same time, the improvement is not caused by more parameters because the variant has the same model size as BERT.
> >
> > > “we adopt widely used 10 probing tasks” — make sure that you actually cite each of the tasks separately, rather than only the benchmark paper that aggregates them.
> >
> > **A13:** We cited relevant work that is used in the benchmark.
> >
> > We cited the two papers you suggested and corrected typos. We hope these explanations can address your concerns and we are always open to discussion.

---

> > > ### Comment · Reviewer_gutD · 2022-11-16
> > > **initial response**
> > >
> > > > TUPE model simplifies Equation 2 by removing two redundant items
> > >
> > > I think you need to rephrase this to clarify what you mean by redundant (in the paper).
> > >
> > > In equation 4, the same variable is redefined twice. What's going on here?
> > >
> > > How did your experimental results change when you removed the context-position interactions from $\delta_{i,j}$?

---

> > > > ### Author Response · Authors · 2022-11-17
> > > > **Clarification of equation 3-5**
> > > >
> > > > Dear Reviewer gutD,
> > > >
> > > > Thanks for your quick response and questions!
> > > >
> > > > > I think you need to rephrase this to clarify what you mean by redundant (in the paper).
> > > >
> > > > **A:** Due to the page limit, we rephrased this part and added more detailed descriptions in Section A of the appendix.
> > > >
> > > > > In equation 4, the same variable is redefined twice. What's going on here?
> > > >
> > > > **A:** It shows two modes for infusing relative position information into attentional weight.  The difference is that the left uses a vector while the right uses a scalar value. We have clarified this in the paper.
> > > >
> > > > >  How did your experimental results change when you removed the context-position interactions from $\delta_{i,j}$?
> > > >
> > > > **A:** Experimental results show the removal of context-position interactions can achieve much better performance on the GLUE benchmark [1] (the results are shown in Table 1 in the original paper).
> > > > We have reproduced this model and obtained similar findings, and the variant with two items removed yields an average improvement of 1.2 points on our 10 sentence-level tasks.
> > > >
> > > > [1] Rethinking Positional Encoding in Language Pre-training
> > > > Guolin Ke, Di He, Tie-Yan Liu
> > > >
> > > > **We look forward to your follow-up response!**

---

> > > > > ### Comment · Reviewer_gutD · 2022-12-06
> > > > > **Thank you for your revisions**
> > > > >
> > > > > You've made substantial improvements! However, I am unsure how novel these results or considerations are in light of the previous work that I had mentioned (AliBi and previous work on locality in LSTMs). I'm updating my score but I am not confident enough to raise it all the way to an accept.

---

> > > > > > ### Author Response · Authors · 2022-12-11
> > > > > > **Thanks for the response!**
> > > > > >
> > > > > > Dear Reviewer gutD,
> > > > > >
> > > > > > Thanks for your response very much!
> > > > > >
> > > > > > ___
> > > > > > > You've made substantial improvements!
> > > > > >
> > > > > > Thanks for your constructive suggestions again.
> > > > > > ___
> > > > > >
> > > > > > > Could you clarify where A1 shows if symmetry is learned vs part of the inductive bias of the position encodings?
> > > > > >
> > > > > > We'd like to clarify this question here detailedly.
> > > > > >
> > > > > > In section 3.2, we discuss the question ***"Is symmetry learned?"***
> > > > > >
> > > > > > To answer this question, we use our proposed metrics, Symmetry, to quantify the positional weight matrix (the averaged weight across layers) before and after pre-training.
> > > > > > Specifically, three language models, *BERT*, *XLNet*  and *DeBERTa* are tested in this experiment, which covers both absolute and relative encodings.
> > > > > >
> > > > > > As shown on the left in Figure 2, the three language models all become
> > > > > > much more symmetrical after pre-training, which proves that the property is indeed
> > > > > > learned. Meanwhile, we observe those human-designed encodings, e.g., sinusoidal and Roformer-like encodings,  satisfy symmetry on purpose.
> > > > > >
> > > > > > Because positional encodings in BERT are randomly initialized, we further explore this question ***"what happens if a language model is initialized with positional
> > > > > > encodings with good locality and symmetry?"***
> > > > > > For this purpose, we replace the positional encodings in BERT with handcrafted Positional Encodings to probe.
> > > > > > Experimental results (as shown in Table 1) demonstrate positional encodings with initialization
> > > > > > of suitable locality and symmetry can have a better inductive bias in sentence representation.
> > > > > > ___
> > > > > >
> > > > > > > I am unsure how novel these results or considerations are in light of the previous work that I had mentioned (AliBi and previous work on locality in LSTMs).
> > > > > >
> > > > > > Thanks for the question.
> > > > > >
> > > > > > Our main contribution is not the new positional encoding, but the insights on how positional encodings work in general:
> > > > > >
> > > > > > 1. Based on your suggestion, we introduced two metrics to quantify the two properties of positional encodings, which shows that the two properties are learned and better locality and symmetry can yield better inductive bias. This is not systematically studied by prior work. Our findings might give an explanation about why ALiBi is designed in that way.
> > > > > > 2. We design two new probing tasks of word swaps, which show a weakness of existing positional encodings, namely the insensitivity against the swap of semantic roles, and this is overlooked by prior work. ALiBi suffers from this issue as well because it meets the symmetry perfectly.
> > > > > > 3. We are the first to probe the linguistic roles of positional encodings
> > > > > > 4. Finally, we propose a novel way to combine positional and contextual encodings, which can bring performance improvement without introducing complexity. Besides, the discussion of ALiBi is included, and we added the ALiBi experiment (see Table 3).
> > > > > >
> > > > > > Overall, we are not proposing a new positional encoding, and we hope our new probing results and findings can shed light on how to design and inject positional encodings into language models.
> > > > > > ___
> > > > > >
> > > > > > We appreciate your useful questions and helpful interactions. Let us know if you have any further questions.

---

> > ### Author Response · Authors · 2022-12-02
> > **A kind reminder**
> >
> > Dear Reviewer gutD,
> >
> > Considering that the deadline for discussion is approaching, we'd like to know if our response has addressed your questions.  We're always happy to answer your further questions.

---

> > ### Comment · Reviewer_gutD · 2022-12-06
> > **A1**
> >
> > Could you clarify where A1 shows if symmetry is learned vs part of the inductive bias of the position encodings?

---

### Official Review · Reviewer_LpaP · 2022-10-24

**Confidence:** 3
**Correctness:** 3
**Technical Novelty And Significance:** 2
**Empirical Novelty And Significance:** 2
**Recommendation:** 5

**Clarity, Quality, Novelty And Reproducibility:**

- The analysis part is novel with good quality.
- The paper is easy to read but missed many details (see weaknesses). Some typos for example BERT->BEET
- The final part of this paper, where a strong position encoding method is proposed, but it was not detailed and analyzed where the improvements are from. This part can be improved by adding another PLM with the proposed position encoding method.

**Strength And Weaknesses:**

Strength:
- Some probing experiments are novel and the findings are interesting
- The proposed methods get good results

Weaknesses:
- Many details are missed, which makes parts of this paper hard to be understood. 1) what are the learnable form of equations 12 and 13? A random initialized matrix? 2) What is PA(X)? Multiply X with the PA matrix? 3) The results of STS and SICK-R seem much better than RoBERTa-large. What is the metric of STS and SICK-R? Spearman or Pearson? STS has 12-16 and STS-B, which one is used in this paper? 4):
- The proposed method is similar to AliBI, in terms of adding a bias mask to the attention scores. Is there any discussion of the difference or which one is better?
- The models are all BERT-base trained with 600K steps. Experimenting with another model can help to support the claims of this paper. For example, show the BERT-large/RoBERTa can also be improved.

**Summary Of The Paper:**

This paper investigates the role of position encoding of pre-trained LM (BERT base) for sentence-level downstream tasks (NLI, STS):
- By visualizing the attention heat map of PLM given a sequence of the same repeating token, this paper finds that LOCALITY and SYMMETRY are two key features of the position encoding of PLM.
- By shuffling spans in different lengths of SNLI data and testing with these shuffled texts, this paper finds PLMs are less sensitive to local word order.

This paper also proposed two handcrafted position encodings, by adding positional bias to the context attention scores.
- By training BERT and BERT with the proposed encoding methods, this paper finds the proposed encoding methods can outperform BERT a lot.

**Summary Of The Review:**

This paper should be improved before it can be published because:

No discussion between Alibi[1] and the proposed method;
Missing details to understand the results;
Only experimented with the BERT base model.




[1] Press, Ofir, Noah Smith, and Mike Lewis. "Train Short, Test Long: Attention with Linear Biases Enables Input Length Extrapolation." International Conference on Learning Representations. 2021.

---

> ### Author Response · Authors · 2022-11-16
> **Reply to Reviewer LpaP**
>
> Dear Reviewer LpaP,
>
> **We would like to thank you for your feedback and useful suggestions on our work!**
>
> > Many details are missed, which makes parts of this paper hard to be understood. 1) what are the learnable form of equations 12 and 13? A random initialized matrix? 2) What is PA(X)? Multiply X with the PA matrix? 3) The results of STS and SICK-R seem much better than RoBERTa-large. What is the metric of STS and SICK-R? Spearman or Pearson? STS has 12-16 and STS-B, which one is used in this paper?
>
> **A1:**
>
> 1) The value $\delta_{i,j}$ is a positional weight between the $i$-th and $j$-th position, which is added to the contextual weight $\gamma_{i,j}$, as described in Equation 5.
> When the encodings are learnable, the $\delta_{i,j}$ in a language model is first initialized by Equation 9. After, $\delta_{i,j}$ is updated during pre-training. We use this experiment to show that encodings with good locality and symmetry can have a better inductive bias. See Section 3.3 for more details.
>
> 2)  PA means positional attention. Equation 5 describes the unified positional encoding. If we only use the $\gamma_{i,j}$, it is the contextual attention (CA). If we only use $\delta_{i,j}$, it becomes Positional Attention (PA).
>
> 3) The metric here is the Spearman correlation, and we use STS-B for evaluation. Here is the STS-B [leaderboard](https://paperswithcode.com/sota/semantic-textual-similarity-on-sts-benchmark) for reference.
>
> We have revised our paper and added all of these explanations in the relevant places.
>
> >  The proposed method is similar to AliBI, in terms of adding a bias mask to the attention scores. Is there any discussion of the difference or which one is better?
>
> **A2:**  There are two key differences between our encodings and other manually designed ones such as the ALibi.
> First, the output generated by our method is an attentional vector (or a discrete probability distribution) that can be regarded as a type of attention mechanism. Thus, we can estimate the locality and symmetry individually. ALiBi biases, in contrast, cannot be measured by our proposed metrics directly.
> Second, we can adjust the hyper-parameters in our method for obtaining encodings with different localities and symmetry, which ALiBi does not allow. Note that our goal is not to outperform ALiBi. Rather, our goal is to design an encoding that allows us to probe and understand the properties of positional encodings in a detailed fashion.
>
> > The models are all BERT-base trained with 600K steps. Experimenting with another model can help to support the claims of this paper. For example, show the BERT-large/RoBERTa can also be improved.
>
> **A3:** We added the experiments for ALiBi, and the results show that our sequence combination can bring improvements to ALiBi as well ( Table 3).
>  |  Model   | Score  |
> |  ----  | ----  |
> | A*  | 81.8 |
> | A*-seq  | 82.5 |
> | AliBi  | 79.7 |
> | AliBi-seq  | 80.4 |
>
> > The final part of this paper, where a strong position encoding method is proposed, but it was not detailed and analyzed where the improvements are from.
>
> **A4:** We enriched this part for better understanding and the implementation example is now on Page 20 (Listing 3).
> We proposed the new method based on the following findings:
> 1)  The main function of positional encodings is to combine local units, e.g., *context* + *##ual* $\rightarrow$ *contextual* and *take* + *off* $\rightarrow$ *take off*.
> 2) Contextual encodings are mainly responsible for long-distance relations (Section 4.2)
> 3) Existing methods all add the two pieces of information together in the attentional head (Equation 5)
> 4)  Intuitively, the composition of local units should happen before the global ones, e.g., the phrase *take off* should be first composed before we put it into a particular context for understanding the meaning.
> Based on these factors, we propose to combine the two encodings in a sequential way.
>
> We hope that these answers can address your concerns and we are always open to discussion.

---

> ### Author Response · Authors · 2022-11-18
> **A kind reminder to Reviewer LpaP**
>
> Dear Reviewer LpaP,
>
> We'd like to kindly remind you that the discussion stage 1 will end soon. Could you have a look at our reply, and we're always happy to answer your further questions.
>
> Look forward to your response. Many thanks!

---

> ### Author Response · Authors · 2022-12-11
> **A kind reminder**
>
> Dear Reviewer LpaP,
>
> The final rebuttal deadline is coming. We have already made revisions to our manuscript and we'd like to know if our updates and reply have addressed your concerns.
>
> We appreciate it very much if you could have a look at our newly updated manuscript and response. Many thanks.

---

### Official Review · Reviewer_eCeS · 2022-10-24

**Confidence:** 4
**Correctness:** 3
**Technical Novelty And Significance:** 2
**Empirical Novelty And Significance:** 2
**Recommendation:** 5

**Clarity, Quality, Novelty And Reproducibility:**

- Similar conclusions have already analyzed by previous work (such as visualizing the positional attentions and handcrafted sine and cosine positional encodings).
- Something unclear to be clarified (see the second point of the weakness)

**Strength And Weaknesses:**

Strength
- Comprehensive studies.

Weaknesses
- There are too many important contents putting in the appendix. For example, the equation of the proposed encodings should be put in the main context.
- Maybe I misunderstood. But why the symmetry is inconsistent with the examples “a man playing an electric guitar on stage” is totally different from “an electric guitar playing a man on stage”? I feel like the symmetry just implies that the attention of the i-th word to the j-th word is almost equally important to the attention of the j-th word and the i-th word. From my point of view, the locality does capture the meaning change in this case.
- In the original Transformer paper, they also design the positional encodings based on sine and cosine. It's satisfied the locality and symmetry properties as well and is reported to have similar performance to the learned positional encodings. So I am not super excited about the experimental results.

Question
- It's a bit interesting that handcrafted + learnable is better than handcrafted and learnable separately. Is it because handcrafted provides better initialization? I suggest to explore this direction more.

**Summary Of The Paper:**

This paper proposes a deep study on positional encoding for transformer models. They visualize the positional attentions for various models and point out that the current learned positional encodings have two important properties: locality and symmetry. Based on the two properties, they design handcrafted positional encodings and show their advantages by conducting experiments.

**Summary Of The Review:**

Something unclear to be clarified and similar conclusions have already analyzed by previous work.

---

> ### Author Response · Authors · 2022-11-16
> **Reply to Reviewer eCeS**
>
> Dear Reviewer eCeS,
>
> **Thank you for your feedback and critical analysis of our work! We are happy to address your questions.**
>
> > There are too many important contents put in the appendix. For example, the equation of the proposed encodings should be put in the main context.
>
> **A1:** We agree. We have now put all important formulas into the main content of the paper.
>
> > Maybe I misunderstood. But why the symmetry is inconsistent with the examples “a man playing an electric guitar on stage” is totally different from “an electric guitar playing a man on stage”? I feel like the symmetry just implies that the attention of the i-th word to the j-th word is almost equally important to the attention of the j-th word and the i-th word. From my point of view, the locality does capture the meaning change in this case.
>
> **A2:** Suppose we have a sentence *"cats eat mice"*  and a positional weight vector for the position of *"eat"*, $[0.25, 0.5,0.25]$, which is generated by a particular encoding. The vector perfectly satisfies locality and symmetry. However, the encoding is insensitive to the change of word orders like *"mice eat cats"*.
>
> >In the original Transformer paper, they also design the positional encodings based on sine and cosine. It's satisfied the locality and symmetry properties as well and is reported to have similar performance to the learned positional encodings. So I am not super excited about the experimental results.
>
> **A3:** Our main contribution is not the new positional encoding, but the insights on how positional encodings work in general. Our new encoding is but a means to that end: We can adjust the hyper-parameters for obtaining encodings with different localities and symmetry. This allows us to learn and report the correlations between the two properties and downstream tasks. Most notably, we can show a potential weakness of symmetry of the existing encodings, and confirm it with two novel probing tasks. Besides, the discussion of ALiBi is included, and we added the ALiBi experiment (see Table 3).
>
>
>
> > It's a bit interesting that handcrafted + learnable is better than handcrafted and learnable separately. Is it because handcrafted provides better initialization? I suggest to explore this direction more.
>
> **A4:** Yes, our experimental results show that BERT initialized with encodings that have good locality and symmetry can have a better inductive bias. We have enriched this part of the paper by introducing two quantitative metrics (Section 3.1) and adding additional experiments (Section 3.2).
>
> We have revised our manuscript based on your comments and we are always open to discussion.

---

> > ### Comment · Reviewer_eCeS · 2022-11-22
> > **Thanks for the response**
> >
> > Thanks for the response. I adjust the score accordingly.

---

> ### Author Response · Authors · 2022-11-18
> **A kind reminder to Reviewer eCeS**
>
> Dear Reviewer eCeS,
>
> We'd like to kindly remind you that the discussion stage 1 will end soon. Could you have a look at our reply, and we're always happy to answer your further questions.
>
> Look forward to your response. Many thanks!

---

### Official Review · Reviewer_WDLJ · 2022-10-25

**Confidence:** 3
**Correctness:** 3
**Technical Novelty And Significance:** 3
**Empirical Novelty And Significance:** 2
**Recommendation:** 6

**Clarity, Quality, Novelty And Reproducibility:**

As mentioned, the quality of writing undervalues the contributions of this paper. Some other aspects that could improve reading
- Some methodology is under-discussed in the main paper. For example, the Identical Word Probing task used in Figure (2) task in never defined in the main text.
- The paper has (in my opinion) an over-extended discussion around anecdotical cases. For example, the “a man playing an electric guitar on stage” is discussed multiple times in different sections of the paper.

**Strength And Weaknesses:**

In terms of strenghs
- The paper has a very throughout analysis of different models and positional encodings
- It proposes several contributions, including a new probing task and several positional encoding methods.

My main problem with this paper at the moment is that it could be much better written: The overall narrative of the paper is unpolished and it’s hard at first to understand the contributions of the paper. I had to go multiple times back-and-forward in this paper to understand what was new in it. The introduction of two different positional encoding methods in different sections is confusing (even if one only serves the purpose of understanding the role of positional encodings). A lot of attention/space is dedicated to locality and symmetry properties of positional encodings, which from my understanding, isn’t very novel and has been explored in previous work.

**Summary Of The Paper:**

This paper studies the role that positional embedding play in large language models through a series of probing tasks. In particular, they first use the previously proposed task of *Identical Word Probing (*where the same word is repeated and passed to a model and the attention weights in the attention-mechanism are visualized) and find that, similarly to previous work, these embeddings seem to pass two properties to the attention weights: symmetry and locality.

They then propose a new probing task, *word swap probing,* where words inside constituents are swapped and the predictions of the model are compared. For example, they observe that swapping word inside noun/verb/etc... phrases as little effect on the predictions, as expected, but that swapping agent and patient also don’t change prediction when it should. They also propose a set of hand-crafted positional encodings that satisfy the locality and symmetry properties and show that they perform comparably to learned kernels.

The authors explore how positional and contextual encodings encode different linguistic features through a series of linguistic probing tasks, finding that positional encodings encode knowledge for lower-level, syntactic probing tasks while on semantic tasks contextual embeddings/encodings perform better. The authors conclude with a new proposed positional encoding mechanism, where-as an attention mechism is applied only to the positional encodings and after another mechanism is applied to the contextual encodings (ie embeddings), finding that it performs comparably or better than a BERT models trained with traditional positional encodings.

**Summary Of The Review:**

This paper provides an extensive analysis of the role of positional embeddings in language encoders. While the analysis is throughout and covers many aspects, and there are some novel contributions in terms of diagnosis tasks and positional encoding methods, the quality of the narrative and writting is sub-par to the point that it's hard to understand the contributions of the paper.
Overall I still think the contributions are novel enough to warrant this conference, and would be willing to increase my score if writing was improved.

UPDATE: The updated manuscript improved the quality of writing massively. There are still some concerns that other reviewers highlighted (such as if these findings generalize to models other than BERT). Overall, raising my score to 6.

---

> ### Author Response · Authors · 2022-11-15
> **Reply to Reviewer WDLJ**
>
> Dear Reviewer WDLJ,
>
> **We thank you for your in-depth review and valuable suggestions, and we are encouraged that you think the contributions of our work are novel enough to warrant this conference.**
>
> > My main problem with this paper at the moment is that it could be much better written.
>
> **A1:** To make our manuscript easy to understand, we revised it from the following aspects:
> * We rewrote the introduction section by emphasizing the differences between our work and prior work and highlighting our contributions in an explicit way
> * We move the preliminary section from the appendix to the main content
> * To better understand the locality and symmetry, two quantitative metrics are introduced. Meanwhile, additional experiments are conducted to analyze the two properties (see Figure 2)
> * We removed some redundant descriptions for simplicity: We moved the linguistic theories of the two properties to the appendix and reduced the references to the running example.
> Besides, some redundant baselines have been removed, e.g., the BERT-A and the identical encoding.
>
> > Some methodology is under-discussed in the main paper.  For example, the Identical Word Probing task used in Figure (2) task is never defined in the main text.
>
> **A2:** We moved some detailed descriptions from the appendix to the content. In particular, we added the definition of the Identical Word Probing task to Section 3.1.
>
> We hope these revisions can make our manuscript clear, and we are always open to further discussion.

---

### Author Response · Authors · 2022-11-16
**A general reply to all reviewers**

Dear reviewers,

**We'd like to thank all reviewers for their time, detailed feedback, and constructive suggestions. The individual reply to each reviewer is provided, and the corresponding revisions in the manuscript are highlighted in blue. In this revised version, we have made the following main changes**:

* We rewrote the introduction section by emphasizing the differences between our work and prior work and highlighting our contributions in an explicit way
* To better understand the locality and symmetry, two quantitative metrics are introduced. Meanwhile, additional experiments are conducted for analyzing the two properties (see Figure 2)
* Important formulas are moved from the appendix to the main content
* The discussion of ALiBi is included, and we added the ALiBi experiment (see Table 3)
* We removed some redundant descriptions and baselines (e.g., BERT-A and the identical encoding) for simplicity

Besides, we added more references based on your comments:

[1] Shortformer: Better Language Modeling using Shorter Inputs

[2] Novel positional encodings to enable tree-based transformers

[3] Transformer Language Models without Positional Encodings Still Learn Positional Information

[4] Sharp Nearby, Fuzzy Far Away: How Neural Language Models Use Context

[5] Aligning books and movies: Towards story-like visual explanations by watching movies and reading books

[6] The LAMBADA dataset: Word prediction requiring a broad discourse context

[7] Fine-grained analysis of sentence embeddings using auxiliary prediction tasks

[8] Does string-based neural MT learn source syntax?

**We hope our replies can address your concerns and we would appreciate it very much if we could have new feedback on our revised paper from your side. We're always happy to answer your questions and further modify our paper.**

---

### Decision · Program_Chairs · 2023-01-20

**Decision:**

Reject

**Justification For Why Not Higher Score:**

Most reviewers think this paper does not have a focused contribution. Although there are many analyses,  the empirical part still does not convince the reviewers.

**Justification For Why Not Lower Score:**

n/a

**Metareview: Summary, Strengths And Weaknesses:**

This paper empirically analyzed the role of position embedding in several NLP tasks. In all, the reviewers admitted that there are comprehensive studies. However, most reviewers think this paper does not have a focused contribution. Although there are many analyses,  the empirical part still does not convince the reviewers.

**Summary Of Ac-Reviewer Meeting:**

n/a